

# The Western United States Large Forest-Fire Stochastic Simulator (WULFFSS) 1.0: A monthly gridded forest-fire model using interpretable statistics

A. Park Williams[1,2], Winslow D. Hansen[3], Caroline S. Juang[4], John T. Abatzoglou[5], Volker C. Radeloff[6], Bowen Wang[7], Jazlynn Hall[3], Jatan Buch[8], Gavin D. Madakumbura[1]

[1]Department of Geography; University of California, Los Angeles; Los Angeles, CA; USA
[2]Department of Atmospheric and Oceanic Sciences; University of California, Los Angeles; Los Angeles, CA; USA
[3]Cary Institute of Ecosystem Studies; Millbrook, NY; USA
[4]Department of Earth and Environmental Sciences; Columbia University; New York, NY; USA
[5]Management of Complex Systems Department; University of California, Merced; Merced, CA; USA
[6]SILVIS Lab; Department of Forest and Wildlife Ecology; University of Wisconsin, Madison; Madison, Wisconsin; USA
[7]Department of Civil and Environmental Engineering, Massachusetts Institute of Technology, Cambridge, MA; USA
[8]Department of Earth and Environmental Engineering; Columbia University; New York, NY; USA

*Correspondence to*: A. Park Williams (williams@geog.ucla.edu)

**Abstract.** We developed WULFFSS, a new stochastic monthly gridded forest-fire model for the western United States (US). Operating at 12-km resolution, WULFFSS calculates monthly probabilities of forest fires ≥100 ha and area burned per fire. The model is forced by variables related to vegetation, topographic, anthropogenic, and climate factors, organized into three indices representing spatial, annual-cycle, and lower frequency temporal domains. These indices can interact, so variables

promoting fire in one domain amplify fire-promoting effects in another. Fire probability and size models use multiple logistic and linear regression, respectively, and can be easily updated as new data or ideas emerge. During its training period of 1985–2024, WULFFSS captures 72% and 83% of observed interannual variability in western US forest-fire frequency and area, respectively. It reproduces regional differences in seasonal timing, frequencies, and sizes of fires, and performs well in cross-validation exercises that test the model's accuracy in years or regions not considered during model training. While lacking

fine-scale fire dynamics, WULFFSS' use of classic statistics promotes interpretability and efficient ensemble generation. Designed to run within a vegetation ecosystem model, bidirectional feedbacks between vegetation and fire can identify how ecosystem changes have altered or will alter fire-climate relationships across the western US. The model's predictive power should improve with increasingly accurate and extensive observational data, and its approach can be extended to other regions. Here we provide a thorough description of the WULFFSS model, including the motivation underlying its development, caveats

to our approach, and areas for future improvement.





## 1 Introduction

In the western United States (US), the annual wildfire area increased by approximately 250% from 1985–2024, largely because annual forest-fire extent increased 10-fold during this time (Fig. 1a). These rapid increases in annual area burned over the last few decades occurred despite ever-intensifying efforts to suppress wildfire (Fig. 1b), signifying a break from the ease with

which fires were contained through most of the 20[th] century. Severe, stand-replacing forest fires also appear to have been more prevalent in recent decades than in previous centuries (Parks and Abatzoglou, 2020; Hagmann et al., 2021; Higuera et al., 2021; Parks et al., 2023; Williams et al., 2023). Thus, even though western US fire are still less common than during pre-European centuries (Parks et al., 2025), the rapid recent increase in fire activity has often not been ecologically restorative (Coop et al., 2020). Further, carbon emissions from increasingly large and severe fires work against carbon-neutrality targets

for climate change mitigation (Anderegg et al., 2022, 2024; Jones et al., 2024). Growing sizes (Juang et al., 2022) and spread rates (Balch et al., 2024) of severe forest fires in the western US have also combined with growing human populations in fire-prone areas (Radeloff et al., 2023) to increasingly put people and property in harm's way (Higuera et al., 2023), including via air pollution far from the flames themselves (Burke et al., 2023). Continued growth in forest-fire sizes and severities may also alter mountain hydrology, with cascading impacts on water resources and flood risk (Kampf et al., 2022; Williams et al., 2022).

These trends motivate improved understanding of, and capability to model, past and future changes to western US forest-fire activity.

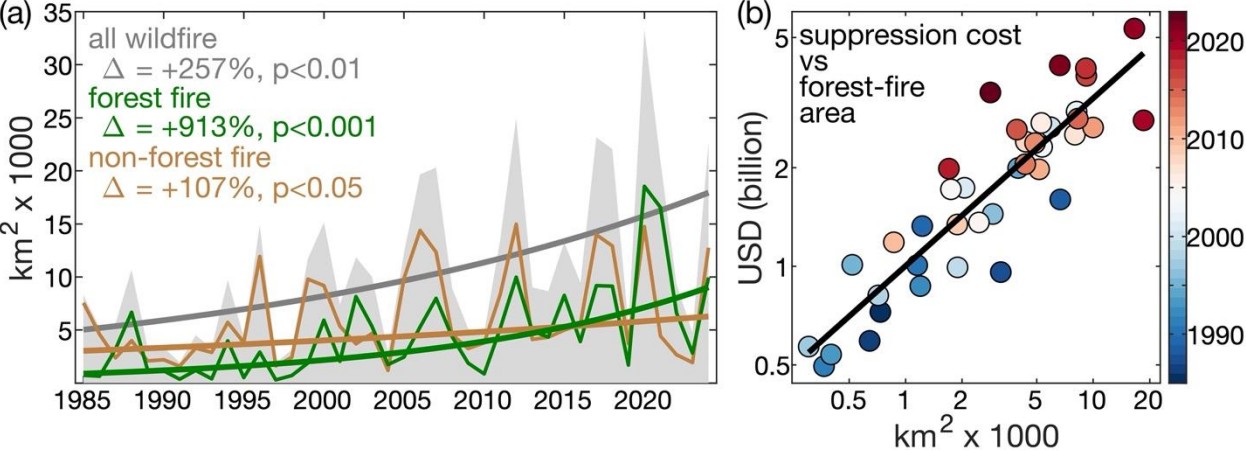

**Figure 1: Annual western US wildfire extent and suppression expenditures.** (a) Time series of annual western US (grey)
total wildfire area, (green) forest area burned, and (brown) non-forest area burned from 1985–2024. Bold trend lines show the Theil-Sen trends in the logarithm of area burned. Delta (Δ) values indicate the relative change in from the first to last year of each trend line and p-values indicate trend significance assessed with one-tailed block (2-year) bootstrap. (b) Scatter plot of annual federal fire suppression cost versus forest-fire area (colors correspond to year) from 1985–2023 (suppression cost unavailable for 2024). Western US refers to the 11 westernmost states in the continental US. Federal suppression costs from
www.nifc.gov/fire-information/statistics/suppression-costs and inflation-adjusted to 2024 US dollars. Fire dataset described in section 3.1.





Drying and warming have been primary drivers of the increase in western US forest area burned in recent decades (Westerling et al., 2006; Abatzoglou and Williams, 2016; Holden et al., 2018; Williams et al., 2019; Brown et al., 2023). Precipitation declines from the early 1980s to the early 2020s were promoted by trends toward the cool states of the El Niño-Southern Oscillation and Pacific Decadal Variability (Lehner et al., 2018), which were probably mostly due to natural climate variability but potentially also promoted by anthropogenic forcing (Hwang et al., 2024; Jiang et al., 2024). The linkage between anthropogenic forcing and warming is clearer and likely to continue (Vose et al., 2017). Warming primarily reduces forest fuel moisture by enhancing the atmosphere's evaporative demand, melting snow earlier in the year, and extending the season of vegetation water use. Temperature drives atmospheric moisture demand through its exponential impact on the vapor pressure deficit ($VPD$), and this variable is strongly correlated with annual forest-fire area in the western US (He et al., 2025) (Fig. 2, left side). Fuel moisture and wildfire activity are also critically affected by other climate variables, including precipitation total, precipitation frequency, and dry windiness (Abatzoglou and Kolden, 2013; Williams et al., 2015; Holden et al., 2018; Brey et al., 2021). Considering a number of methods to quantify fuel aridity, Abatzoglou and Williams (2016) attributed approximately half of the western US forest area burned from 1984–2015 to anthropogenic climate trends. However, that study's analysis was not spatially explicit, it focused exclusively on area burned, and it did not consider contributions from other human impacts on fire, such as through land use, fire suppression, or ignitions.

Fuel characteristics are also key determinants of wildfire activity, in part because they modulate the sensitivity of fire to climate (Bradstock, 2010; Littell et al., 2018). As long there are sufficient lightning or human ignitions, increased abundance and connectivity of flammable fuels will make fire activity more responsive to aridity (Fig. 2). In non-forested areas of the western US, where fuels are generally more limiting due to less biomass and connectivity, the relationship between area burned and $VPD$ is considerably weaker than in forested areas (Fig. 2) despite non-forest areas on average being warmer, drier, and therefore more likely to burn based on fuel moisture alone.

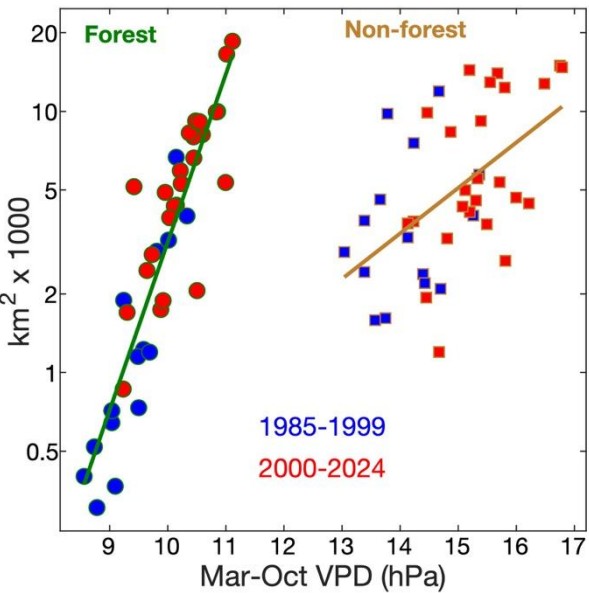

**Figure 2: Annual wildfire area versus atmospheric aridity.** Regressions are shown for forested (circles with green outlines) and non-forested (squares with brown outlines) areas of the western US. The vapor-pressure deficit (*VPD*) is a measure of the aridity of the atmosphere, which is a strong determinant of the moisture content of fine fuels such as dead grass, and March–
October (Mar-Oct) is a time period when *VPD* is particularly strongly correlated with annual area burned. Fire and climate data described in sections 3.1 and 3.3.

Fuel characteristics also modulate how fire responds to climate *within* forests, and thus fire activity in a given region and time period may be strongly affected by that region's fire history. In a metanalysis of >1,000 western US forest fires, Parks et al.
(2015) found a self-regulating effect of fire, where fuel reductions caused by past fires tended to limit subsequent fire spread for 5–20 years. In other metanalyses, Parks et al. (2018a) and Hakkenberg et al. (2024) found that pre-fire fuel abundance, and ladder fuels in particular, strongly affect fire severity.

The US practice of fire exclusion has led to artificially high levels of vegetation biomass, spatial continuity, and understory
vegetation in many western US forests (Hagmann et al., 2021). This has been especially detrimental for semi-arid forests where pre-European fire frequencies were on the order of 5–30 years (Swetnam, 1993; Swetnam and Baisan, 1996; Van de Water and Safford, 2011). In these forests, a century or more of little-to-no fire represents a dramatic departure from a historical fire regime typified by frequent, low-intensity surface fires. Resultant fuel accumulation has been conducive to vertical movement of fire into forest canopies (Steel et al., 2015; Hagmann et al., 2021). Accordingly, in many semi-arid western US forests, fire
exclusion is partly responsible for the strength of the positive response of annual forest-fire area to warming and drying.

Importantly, humans are responsible for the vast majority of wildfire ignitions in the US (Balch et al., 2017), so it may be hypothesized that increased population has driven the increase in western US forest-fire activity. According to Syphard et al. (2025), however, the annual frequency of wildfires on US Forest Service lands has declined since approximately 1980. A lack



of increasing fire frequency despite growing population, drier fuels, and, in many forest areas, greater fuel abundance, is consistent with a globally observed tendency for increased human population to have a net-negative effect on fire frequency in all but the most sparsely populated regions (Bowman et al., 2011; Knorr et al., 2014). Thus, the rapid increase in western US forest-fire extent and associated impacts has been driven by growing fire sizes, not ignition frequency (Juang et al., 2022).

The effects of continued changes in climate, fuel characteristics, and human activities on western US forest-fire frequency, extent, and other critical metrics such as severity and carbon/smoke will be complex and heterogeneous across space and time (Williams and Abatzoglou, 2016; Littell et al., 2018; Buotte et al., 2019). This heterogeneity will be driven in part by ecosystem dynamics. That is, if changes in climate and/or human activities cause large changes in fire activity, they will affect forest ecosystems and fuel characteristics such as aboveground biomass, connectivity, and stand age, and these ecosystem changes
will in turn affect fire.

A continued rapid increase in forest-fire area may become self-regulating as fuel loads and connectivity decline (Parks et al., 2015, 2018b), but forecasting the timing, magnitude, and geography of this effect requires understanding of complex fire-induced mortality and succession (Harvey et al., 2016). In simulations with the LANDIS-II model, Hurteau et al. (2019) found
that both coupled and uncoupled simulations resulted in large increases in area burned and fire emissions, but that the coupled simulations had a small self-regulating effect that reduced projected trends by 10–15%. However, LANDIS-II is computationally intensive and this study was confined to three representative transects within the Sierra Nevada, rather than the whole Sierra Nevada. In addition, Hurteau et al. (2019) made simplifying assumptions that fire ignitions are randomly distributed across the landscape and fire effects on biomass only last for 10 years. Taking a much simpler approach, Abatzoglou
et al. (2021) performed simulations treating the entire western US forest area as essentially a single model grid cell to assess how sensitive future western US forest-fire trends are to assumptions about the strength of fire's self-regulating effect. Interestingly, even simulations that assumed a very strong self-regulating effect projected continued rapid increases in annual forest fire area, though at only half the rate as simulations assuming no self-regulation. In addition to not considering spatial variability, Abatzoglou et al. (2021) focused solely on area burned and the simulations lacked ecological dynamics. As such,
they modeled only until 2050 and did not assess whether the self-regulating effect of larger fires may be more pronounced for other variables such as fire intensity, severity, or biomass combusted.

Most wildfire impacts are caused by a relatively small number fires (Moritz et al., 2005). In the western US, approximately 90% of the total area burned is accounted for by fewer than 10% of wildfires according to the US Forest Service's Fire Program
Analysis Fire-Occurrence Database (FPA-FOD) (Short, 2022). Given that larger fires tend to burn at higher severity (Cova et al., 2023), realistic simulation of future fire-vegetation coupling requires modeling extreme fire events. For realistic simulations of complex processes, a mechanistic modeling approach that explicitly simulates fine-scale processes such as combustion and energy transfer is ideal. However, the temporal and spatial scales at which fine-scale mechanistic fire models



can be run are severely limited by computational constraints. For example, coupled atmosphere/fire models such as
HIGRAD/FIRETEC (Linn, 1997; Linn et al., 2012), CAWFE (Coen, 2013) and WRF-Fire (Muñoz-Esparza et al., 2018) can
only feasibly operate at a scale of tens of kilometers at most, insufficient to understand the drivers of historical and future
wildfire activity across the large scale of the western US. One model designed for efficient simulation of fire dynamics across
regions as large as the western US is SPITFIRE (Thonicke et al., 2010; Lasslop et al., 2014), which is described as process-
based because is simulates fire intensity and wind-driven fire spread following Rothermel's equations (Rothermel, 1972;
Andrews, 2018). However, the rules that govern ignitions and whether fuels are abundant and dry enough to burn are
empirically parameterized. An advantage of mechanistic, or process-based, models is that they are deterministic; a given set
of predictor conditions will always lead to the same fire outcome, making them diagnosable and replicable. Their disadvantage
is that at the relatively low spatial and temporal resolutions necessary for decadal to centennial simulations across a large
region like the western US, a model like SPITFIRE is likely to underrepresent variability and extremes.


Due to the limitations of all other forest-fire models, we developed a new stochastic monthly forest-fire model, WULFFSS,
which is a gridded statistical forest-fire model for the western US. We designed this model to operate in a coupled framework
within a forest ecosystem model, the Dynamic Temperate and Boreal Fire and Forest-Ecosystem Simulator (DYNAFFOREST)
(Hansen et al., 2022). The WULFFSS model simulates the monthly occurrences and sizes of forest fires $\geq 1$ km$^2$ in size on a
12-km resolution grid. Fire probabilities and sizes are determined as functions of fuel characteristics, topography, human
population, and climate/weather. WULFFSS reproduces realistic spatiotemporal variations in fire frequency and area burned
under historical conditions, and its use of conventional statistics promotes interpretability of model behavior and outputs. The
model's computational efficiency and stochastic nature allow for many simulations of monthly forest fire activity across the
western US for decades or centuries at a time. Implementation of WULFFSS within a forest ecosystem-model such as
DYAFFOREST will allow for simulation of the coupled interactions between fire and ecosystem dynamics that will ultimately
shape how the western US forest-fire regime evolves under anthropogenic climate change. While WULFFSS was built to be
coupled with DYNAFFOREST, it is designed in a modular fashion where coupling with other vegetation models should be
relatively straight forward.

**2 Geographic domain**

Our study area is the forested domain of the eleven westernmost states of the coterminous US: Arizona, California, Colorado,
Idaho, Montana, New Mexico, Nevada, Oregon, Utah, Washington, and Wyoming. Consistent with other work in the region
(Buotte et al., 2019; Hansen et al., 2022), we determine the forested domain from the 250-m forest map from Ruefenacht et al.
(2008), from which we calculate a 1-km resolution map of fractional forest coverage. We classify a given 1-km grid cell as
forested if $\geq 50\%$ of the 250-m grid cells are forest. From this 1-km forest map, we determine our 12-km resolution model
domain to include all 12-km grid cells containing at least one forested 1-km grid cell. We remove 12-km grid cells immediately



south of the Canadian border because some of our landcover- and population-related predictor variables require information from surrounding grid cells beyond the US. In total, there are 11,132 12-km grid cells within our forested study domain of the western US (Fig. 3). In assessments of regional model performance we consider the four quadrant regions mapped in Fig. 3: Pacific Northwest (PNW), Northern Rockies (N Rockies), California and Nevada (CA/NV), and the four-corner states (4 175    Corners).

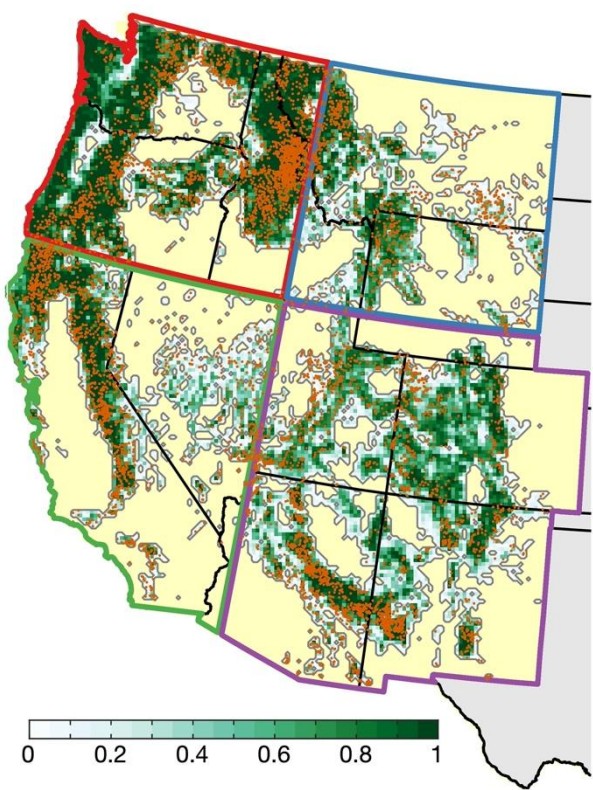

**Figure 3: The western US study domain.** Grey contour outlines the western US forested study region. Shades from white to green: fractional forest cover in each 12-km grid cell within the forested study region according to the Ruefenacht et al. (2008) 180    forest map. Orange dots: ignition locations of forest fires ≥100 ha in the study region from 1985–2024. Yellow: non-forested areas of the western US. Grey: outside the western US. Colored boundaries identify the four quadrant regions considered in regional analyses: Pacific Northwest (red, PNW), Northern Rockies (blue, N Rockies), California and Nevada (green, CA/NV), and the four-corner states (purple, 4 Corners).

## 3 Data

### 3.1 Forest fire



To parameterize the fire model we use the Western US MTBS-Interagency (WUMI2024a) database of observed wildfires from 1984–2024 (Williams et al., 2025). Like its predecessor described by Juang et al. (2022), the WUMI2024a was developed by harmonizing several public US government sources and it does not include fires <1 km² in size. The WUMI2024a contains a list of western US wildfire events, including ignition date, ignition location, and final fire size, as well as a 1-km resolution map of the area burned by each fire. For wildfires represented in the interagency Monitoring Trends in Burn Severity dataset (MTBS) (Eidenshink et al., 2007), burned-area maps and fire sizes are calculated from MTBS's 30-m resolution satellite-based maps of fire-severity classifications, thus accounting for unburned islands within fire perimeters. See Williams et al. (2025) for details about the data sources and the methods used to develop the WUMI2024a. We constrain calibration of the WULFFSS to 1985–2024 due to a suspicious complete absence of fires from Wyoming and New Mexico in 1984.

We estimate forest area burned by each fire in the WUMI2024a and only retain fires that burned ≥1 km² of forest area. To estimate the forest area burned by each fire, we multiply each 1-km grid cell of fractional area burned by the fractional forest area and sum. Of the 21,799 wildfires represented in the WUMI2024a in 1985–2024, 7,702 have ≥1 km² forest area burned. However, a number of wildfires are identified in the WUMI2024a as 'parent fires' composed of smaller sub-fires. This occurs because, although the most accurate dataset feeding into the WUMI2024a is the MTBS, that dataset sometimes attributes burned areas from multiple fires to a single event. The WUMI2024a notes these cases, and we replace parent fires with their associated sub-fires. To keep burned areas consistent with the high-quality calculations from MTBS, we scale the forest area burned by each set of sub-fires to sum the parent fire's value. In cases where a sub-fire's ignition location is not within a 1-km forested grid cell that burned, we reassign the ignition location to the nearest grid cell with forest area that burned. We find 56 parent fires composed of at least two sub-fires with ≥1 km² forest area burned after rescaling. After replacing parent wildfires with their sub-fires, our dataset consists of 7,840 wildfires with ≥1 km² forest area burned from 1985–2024. This number is reduced to 7,673 after removing fires ignited in areas outside our western US study domain shown in Fig. 3 because they ignited near the Canadian or Mexican border or in a 12-km grid cell containing no 1-km grids with ≥50% forest area.

## 3.2 Topography

We calculate topographic predictors from the 1-km digital elevation model produced by the NOAA GLOBE project (Hastings and Dunbar, 1998). From the 1-km grid of mean elevation we calculate 1-km grids of slope and aspect. We then calculate 12-km grids of mean slope to represent steepness as well as the standard deviation of 1-km elevation values to represent terrain ruggedness.

## 3.3 Climate

### 3.3.1 Daily 1/24° gridded climate





We calculate climate predictors from daily gridded climate data with 1/24° (~4 km) geographic resolution for January 1951 –
December 2024. This period begins in 1951 rather than coincident with our 1985–2024 study period because the longer climate
record is used to spin-up our forest simulations (section 3.4). Daily variables are precipitation total (*prec*, mm), maximum
temperature (*tmax*, °C), minimum temperature (*tmin*, °C), vapor pressure (*ea*, hPa), mean downwelling solar radiation at the

surface (*solar*, W m$^{-2}$), and mean 2-m wind speed (*wind*, m s$^{-1}$). For *prec*, *tmax*, and *tmin*, data come from the 1/24°-resolution
nClimGrid Daily dataset produced by the National Oceanic and Atmospheric Administration (Durre et al., 2022), which covers
1951–present. For *ea* we apply the Clausius Clapeyron formulation to the daily 1/24°-resolution dew point (*tdew*, °C) dataset
from the PRISM group at Oregon State University (Daly et al., 2021). This dataset is better than reanalysis products because
it is based on station observations. However, the daily PRISM dataset starts in 1981. For 1951–1980, we use a dynamically-

downscaled version of the ERA5 reanalysis for the western US (Rahimi et al., 2022). This reanalysis has 9-km spatial
resolution and covers September 1950 – August 2023. We use daily outputs of mean specific humidity (*q*, m$^3$ m$^{-3}$) and surface
pressure (*p*, hPa) to estimate *ea*: $pq/(0.622+0.378q)$. We then bilinearly interpolate to 1/24° resolution and use quantile
mapping to bias-correct the Rahimi et al. data such that, for each grid cell and each of the 12 months, the distributions of daily
*ea* estimated from Rahimi et al. match those estimated from PRISM during their January 1981 – August 2023 period of overlap.

For *solar* and *wind*, we prioritize the daily outputs from Rahimi et al. because there are no long-term spatially continuous
records of direct observations of these variables and the Rahimi et al. data have uniquely high spatial resolution and long
temporal coverage. We downscale the Rahimi et al. *solar* and *wind* data to 1/24° resolution using bilinear interpolation.
Because the Rahimi et al. dataset ends in August 2023 we extend through 2024 by bias correcting the daily *solar* and *wind*
data from gridMET (Abatzoglou, 2013). The gridMET product has 1/24° resolution, but for solar and wind this is achieved by

downscaling outputs from the much lower-resolution (~1/3°) North American Regional Reanalysis (NARR) (Mesinger et al.,
2006). Bias-correction of gridMET *solar* and *wind* is once again performed with quantile mapping such that, for each grid cell
and each of the 12 months, the bias-corrected distribution matches that of Rahimi et al. during the post-1979 overlap.

For solar, we account for the effect of slope and aspect on incident solar angle (e.g., solar intensity is higher on south-facing

slopes). The Rahimi et al. reanalysis accounts for the effect of elevation on solar intensity, but not the effect of slope and
aspect. To do this, we use 1-km resolution maps of slope and aspect, calculated from the 1-km maps of mean elevation from
NOAA GLOBE (Hastings and Dunbar, 1998). Our method is to, for each day in a generic 365-day year and assuming a top-
of-atmosphere solar constant of 1367 W m$^{-2}$, use the method developed by Kumar er al. (1997) to estimate the mean
downwelling solar intensity at the surface at 1-km resolution for two scenarios: one with observed elevation, slope, and aspect

(*solar_topo*) and another with observed elevation but assuming topography within each 1-km grid cell is flat (*solar_flat*). For
each day we then calculate an adjustment factor representing the fractional effect of slope and aspect on incident solar radiation
at the surface as *solar_adj = solar_topo/solar_flat*. We then upscale the daily grids of *solar_adj* to 1/24° resolution and



calculate a topography-adjusted version of *solar* (*solar_topo*) by multiplying each daily map of *solar* by its corresponding map of *solar_adj*.


We use the 1/24° daily climate maps described above to calculate a number of fire-relevant derived variables. We calculate daily mean *VPD* as the average of the daily maximum and minimum *VPD* (*VPDmax* and *VPDmin*, respectively), where *VPDmax* is calculated as the saturation vapor pressure (*es*) at *tmax* minus *ea* and *VPDmin* is calculated using *es* at *tmin*. As a metric representing daily atmospheric fire weather, we use a modified version of the hot-dry-windy index (*HDWI*, hPa m s$^{-1}$)

representing surface conditions. The standard formulation of the *HDWI* (Srock et al., 2018) multiplies *wind* by *VPD* at multiple vertical levels within the bottom 500 m of the atmosphere on a sub-daily time scale (e.g., 6 hourly), and then defines each day's *HDWI* value as the maximum among all values from at any vertical level or time step. Our simplified approach is to estimate daily *HDWI* as *VPDmax* multiplied by *wind*.

To represent the effect of snowpack we use the 4-km daily gridded climate data to simulate daily mean snow-water equivalent (*SWE*, mm) using the SnowClim model (Lute et al., 2022), which is designed for efficient simulation of western US snow dynamics in response to gridded forcing data at a daily or sub-daily time step.

To represent fuel moisture we calculate the daily 100- and 1,000-hour dead fuel moisture content (*FM100* and *FM1000*,
respectively, %) following the method of the National Fire Danger Rating System (NFDRS) (Cohen and Deeming, 1985). The 100-hour and 1,000-hour fuel classes represent woody fuels 25–76 mm and 76–203 mm in diameter, respectively, and the names of the fuel classes represent the approximate e-folding time required for moisture content to equilibrate with the atmosphere. We include the effect of simulated snow in our calculations by setting relative humidity to 100% when the snow depth is ≥5 mm and by withholding precipitation that increases the water content of the snowpack until it melts out of the
snowpack.

### 3.3.2 Monthly 12-km climate predictors

We calculate nearly all monthly climate predictors from the daily 1/24° grids described above. In addition to monthly means we also consider variables representing fire-relevant sub-monthly quantities (e.g., maximum 1- or 3-day mean *HDWI* or *VPD,* maximum single-day *SWE* of the past 12 months) as well as variables representing the integration of climate conditions over
multiple months (e.g., 3-, 6-, 9-, or 12-month mean *VPD*).

In addition to 12-km climate predictors derived from our daily 1/24° dataset, we also consider lightning frequency using the 0.1°-resolution daily maps of lightning-strike density from the National Lightning Detection Network (NLDN, https://www.ncei.noaa.gov/pub/data/swdi/). This dataset begins in 1987 and we aggregate to monthly maps of 12-km lightning



frequency for 1987–2024. However, NLDN methodology changed over time so we only use maps of long-term and monthly climatological mean lightning frequencies as predictors. To account for temporal variability in lightning potential on interannual timescales, we consider monthly mean convective available potential energy (*CAPE*) as well as maximum 1- and 3-day mean *CAPE* from Rahimi et al. (2022), which we upscale to 12-km resolution using bilinear interpolation. Because the Rahimi et al. dataset ends in August 2023, we extend through 2024 by downscaling daily *CAPE* from NARR and bias

correcting such that monthly climatologies of mean CAPE and maximum 1- and 3-day CAPD match Rahimi et al. (2022) in terms of mean and standard deviation during the post-1979 overlap.

**3.4 Landcover**

Due to a lack of spatially continuous and temporally evolving observational maps of fire-relevant forest biomass variables throughout our study period, we simulate forest biomass during our study period using the Dynamic Temperate and Boreal

Fire and Forest-Ecosystem Simulator (DYNAFFOREST) (Hansen et al., 2022). DYNAFFOREST is a process-based forest ecosystem model designed to efficiently simulate forest dynamics across the western US at a medium spatial resolution (grid cell size of 1 km$^2$). The model represents 11 forest types and one grass/shrub type, runs at an annual time step, and simulates a suite of variables representing various stand structure characteristics and ecosystem functions. DYNAFFOREST is a cohort based model; in each forested 1-km grid cell, a single tree representing one forest type is simulated. Simulated metrics from

the single tree are then used to estimate stand structural characteristics for each grid-year, such as stand age, density, basal area, mean canopy height, and diameter at 1.35 m above the ground. DYNAFFOREST tracks 3 live and 3 dead above-ground biomass pools: stem, branch, and foliage, and standing snags, downed coarse wood, and forest floor litter. Cohort mortality occurs probabilistically as a function of background causes, drought, and fire.

When a fire occurs, DYNAFFOREST estimates percent crown kill of the cohort as a function of fuel aridity, tree size, and forest-type specific crown dimensions. Probability of mortality is estimated as a function of crown kill and bark thickness. Following a fire, forest establishment and recovery is simulated in DYNAFFOREST probabilistically based on the fecundity of the surrounding forest types, dispersal distance in the target grid cell and surrounding grid cells, and the effects of climate on seed germination and establishment. Key functional traits related to postfire recovery, like cone serotiny and asexual

resprouting, are included. If stand-replacing fire occurs and postfire establishment does not occur the next year, then the landcover is assumed to convert to grass/shrub, though forest can return if or when seed supplies and climate conditions allow.

Because DYNAFFOREST outputs are not observational, our empirically parameterized fire model will not perfectly represent how observed forest characteristics affect the probabilities and sizes of forest fires. However, DYNAFFOREST has been well

benchmarked across large diverse forest types of the western US (Hansen et al., 2022) and used to simulate coupled fire-forest relations in the context of fuels management (Daum et al., 2024). Additionally, we find reasonable representation of



ecoregional differences in most above-ground biomass pools when we compare DYNAFFOREST outputs with the US Forest Service's Forest Inventory and Analysis survey data (USDA Forest Service, 2019). Further, in the DYNAFFOREST simulation used to produce the 1985–2024 forest maps that we use to parameterize the fire model, we apply the observed 1-km maps of

forest area burned from WUMI2024a. By allowing DYNAFFOREST to simulate forest responses to known fires, our parameterization reflects not just the effects of naturally occurring, long lasting gradients in forest condition on fire, but also more transient, sharper gradients caused by prior fires.

To assure realistic and stable forest dynamics leading into the 1985–2024 parameterization period, we conduct a >334-year

spin-up using WULFFSS coupled with DYNAFFOREST. For the first 300 years (1651-1950), we force DYNAFFOREST with detrended climate data from 1901-1950 and climate years are randomly selected with replacement. For 1951–1984, we observed climate so that forest condition in the WULFFSS parameterization can reflect the legacies of recent climate variations. With the exception of the variables used to force WULFFSS, the climate variables used by DYNAFFOREST are mean June–August 0–100 cm soil moisture and annual forest-type specific temperature metrics such as growing-degree days

and freezing-degree days. Monthly 0–100 cm moisture is modeled from monthly 12-km climate data from 1901–2024 following Williams et al. (2017, 2020) and bilinearly interpolated to 1-km resolution. The temperature metrics are calculated from monthly 1/24° grids of mean of *tmax* and *tmin*. We downscale the 1/24° grids to 1-km resolution guided by the TopoWx dataset (Oyler et al., 2015). Specifically, TopoWx provides monthly grids of *tmax* and *tmin* from 1948–2016 with resolutions of 1/24° and 1/120° (~800 m). For each month and variable, we use the 1/120° (~800 m) version to calculate a mean 1980–

2016 climatology with 1-km resolution (estimating 1-km values from the 1/120° grid using nearest-neighbor interpolation) and then produce a 1-km map of offsets that relate each 1-km climatological mean value to its overlying 1/24°-resolution value from the same years. We apply the offsets to the monthly mean *tmax* and *tmin* from NOAA nClimGrid (Vose et al., 2014) to produce 1-km maps of monthly mean *tmax* and *tmin* from 1901–2024. Thus, we force the non-fire portion of the DYNAFFOREST simulations with observed climate data for the 1901–2024 period.


Due to lack of fine-scale data on forest ecosystems from the pre-spin-up period, we initialize the spin-up using a 1-km resolution map of observed modern forest types that we derived from the 250-m map of Ruefenacht et al. (2008) forest types, following Buotte et al. (2019). Initial fuel loads are representative of the 11 forest types and the biomass pools stabilize after approximately 250 years of spin-up.


For landcover variables not simulated by DYNAFFOREST, we use the maps of land-cover type from the US Geological Survey's National Land Cover Database (NLCD; https://www.usgs.gov/centers/eros/science/annual-national-land-cover-database). The NLCD provides annual maps of landcover classifications at 30-m resolution across the US for 1985–2023. Because these the NLCD map for a given year often reflects the effects of fires during that, and we do not wish to mistake the



effects of fires for their causes, we shift the year represented by each map forward by a year (for 1951–1985 we assign the 1986 landcover). From these 30-m maps of landcover we calculate 1-km maps of fractional coverage for four non-forest landcover categories: unburnable (water, ice, wetland, barren), developed (low, medium, high intensity), agriculture (cultivated, pasture, developed open space), and grass/shrub (grass/herb, shrub/scrub). For each year from 1951–2024 we then rescale these fractional coverages so that, for grid-years where the DYNAFFOREST simulation does not indicate forest coverage, these non-forest classes sum to full coverage. Likewise, for grid-years where DYNAFFOREST simulates forest coverage, we set the non-forest types to zero.

In addition to 1-km maps of aboveground forest biomass density (in distinct pools and in total), mean canopy height, mean diameter at breast height, and fractional coverage by landcover type, we also calculate 1-km maps of forest connectivity. We define this as, for each 1-km grid cell, the fraction of adjoining grid cells with ≥10,000 kg ha$^{-1}$ live biomass density, which corresponds to approximately the 5$^{th}$ percentile of all simulated 1-km$^2$ live biomass density values for 1985–2024. Specifically, for each 1-km grid cell with ≥10,000 kg ha$^{-1}$ live biomass we calculate the number of consecutive adjoining grid cells in each of the 8 directions radiating away from central grid cell that also have ≥10,000 kg ha$^{-1}$ live biomass. In each of the four directions radiating north, south, east, and west, we consider the 6 nearest grid cells. In each of the four diagonal directions we consider the nearest 4 grid cells. We then calculate connectivity as 1 (for the central grid cell) plus the sum of the total number of adjoined grid cells with ≥10,000 kg ha$^{-1}$ grid cells the 8 directions divided by the number of grid cells considered (41).

From the 1-km grids of annual forest properties and fractional coverage by landcover type described above we calculate 12-km maps of averages within each 12-km grid cell. Given that fire sizes can also be influenced by landcover beyond the ignition location, we also consider variables that represent spatial averages within the area of a very large 500 km$^2$ (50,000 ha) fire, which we approximate as a 23 x 23 km square.

**3.5 Human population and roads**

Humans influence western US forest fire directly causing approximately half of all ignitions (Balch et al., 2017) and suppressing almost all wildfires. We therefore include predictor variables related to population density and distance to populated areas. Because the US Census changed how it provides population information in 2020, so that reported numbers are sometimes swapped among Census units ('blocks') to maintain confidentiality, we work with census-based housing-unit density instead. Specifically, we use the shapefiles of census-based, block-level housing density in 2000, 2010, and 2020 developed by the SILVIS Lab (https://silvis.forest.wisc.edu/data/wui-change/) (Radeloff et al., 2018, 2023). For 1950–1990 we use decadal hindcast maps of housing density produced by the SILVIS Lab using partial block-group level census data. For 2030, which is used with 2020 to interpolate housing density for 2021–2024, we use a projection based on county-level forecasts of housing density from Woods & Poole Economics (https://www.woodsandpoole.com/our-databases/united-states/),





which is downscaled to the block level by the SILVIS Lab based on 2020 housing density patterns. For each decade we rasterize the polygon data to a 1-km grid of housing density. We then produce annual maps of 1-km housing density for 1951–2024 by linearly interpolating between the decadal maps.


From the annual 1-km maps of housing density we produce two sets of 1-km maps to represent distance from populated areas. In the first, we map the distance to the nearest grid cell with $\geq 5$ housing units km$^{-2}$ to represent distance to a relatively sparsely populated community. In the second we map the distance to the nearest grid cell with $\geq 50$ housing units km$^{-2}$ to represent distance from an urban center.


The geographic distribution of ignitions and fire-suppression activities also depend on roads. We use the 2013 Global Roads Open Access Data Set, Version 1 (gROADSv1; https://search.earthdata.nasa.gov/search/granules?p=C1000000202-SEDAC. This dataset specifies for each road segment a Functional Class: Highway, Primary, Secondary, Tertiary, Local/Urban, Trail, Private, or Unspecified. We aggregate these into two classes: major (Highway and Primary) and minor roads (all others). We

then produce 1-km maps of the distance to the nearest major road, distance to nearest minor road, and distance to nearest road of any class. We treat the road network as static in time due to unavailability of construction or closure dates.

Finally, we calculate 12-km maps of mean 1-km housing density, distance to nearest location with $\geq 5$ or $\geq 50$ housing units km$^{-2}$, and distance to nearest major road, minor road, or any road as predictor variables in the forest-fire model.

**4 Model description**

The WULFFSS model has a spatial resolution of 12 km across the forested domain of the western US (Fig. 3) and operates monthly. The model is parameterized on the dataset of 7,673 forest-fire locations and sizes described in section 3.1. WULFFSS consists of three statistical models, loosely following Westerling et al. (2011). The general framework is that first model estimates, for each grid-month, the probability of $\geq 1$ wildfire ($P$) from a multi-variate logistic regression with predictor

variables representing landcover, topography, humans, and climate. To account for the possibility of $>1$ wildfire in a given grid-month, the second model then uses $P$ as a single predictor in a logistic regression to estimate the number of wildfires in each grid-month ($N$). The third model is a fire-size model that uses multi-variate regression to estimate the forest area burned ($A$) by each wildfire as a function of landcover, topography, humans, and climate, similar to the $P$ model.

The $P$ and $A$ models each consist of three components representing spatial variations ($S$), the mean annual cycle ($C$), and temporal anomalies ($T$), as well as interactions between these components ($SC$, $ST$, and $CT$). The $S$ component represents drivers of forest-fire occurrence or size that are most variable in the spatial domain, such as topographic slope, fuel availability, human population, mean annual lightning frequency, and long-term mean aridity, all of which may directly influence fire



occurrence and also modulate the effects of *C* and *T*. Each potential *S* predictor is standardized such that all grid-month values

in the study domain have a mean of 0 and standard deviation of 1 for the calibration period (1985–2024). Many *S* predictors represent alternate expressions of a single predictor, for example house density, $\log_{10}$(house density), mean house density within 50 kha, and $\log_{10}$(mean house density within 50 kha).

The *C* component represents drivers of forest-fire occurrence or size that are most variable in the domain of the mean annual

cycle, such as long-term means of each month's lightning frequency as well as variables that influence the seasonality of fuel moisture such as *prec*, *solar*, and *VPD*. For all potential *C* predictors, mean annual cycles are calculated for the calibration period. As for *S*, most potential *C* variables are permutations of common variables. For example, the effects of climate variables related to fuel moisture may accumulate over several months, so the annual cycle of each climate variable is considered as 1-, 2-, 3-, 4-, and 5-month running means. Further, two versions of most *C* variables are considered. In the first, each grid cell's

mean annual cycle is scaled from 0–1, where 0 and 1 represent the mean annual minimum and maximum, respectively, so all spatial variability is due to variability in the timing of the annual cycle. In the second, mean annual cycles are not scaled and these variables retain spatial differences in each month's mean conditions. For each of these unscaled *C* variables, values are standardized relative to all calibration-period grid-months.

The *T* component represents drivers of forest-fire occurrence or size that are most variable in the temporal domain of interannual and longer. Potential *T* predictors include the standardized precipitation index (*SPI*) (McKee et al., 1993), frequency of wet days with ≥2.54 mm *prec* (Holden et al., 2018), *VPD*, *solar*, *HDWI*, *CAPE*, and *SWE*. As for *C*, many potential *T* variables are permutations to represent cumulative effects over various ranges of months. In addition, monthly measures of some sub-monthly meteorological conditions are considered such as the highest 1-, 3-, or 5-day mean *VPD* within

a month. Because *T* is meant to represent climate variability beyond the annual cycle, *T* variables are standardized so that for a given variable in a given grid cell, values have a mean of 0 and standard deviation of 1 for each of the 12 months during the calibration period.

In both the *P* and *A* models, each of the three components is represented by a single composite index that is an expression of

the combined effect of multiple predictor variables. The variables that contribute to each of the three components (*S*, *C*, and *T*) are selected stepwise and only retained if they contribute significantly to model skill (see section 4.1). Thus, each model ultimately uses only a subset of the potential predictors. Lists of all potential *S*, *C*, and *T* predictors considered for the models are listed in Tables A1–A3 (see Tables S1–S3 for variable descriptions). For some variables, it is logical that the effect on *P* or *A* should be only positive or negative. For example, the direct effect of fuel availability on fire occurrence and size is far

more likely to be positive than negative, but a statistical model may detect a hump-shaped or even negative relationship due to the co-occurring influences of moisture on fuel availability (positive) and flammability (negative) (Bradstock, 2010; Krawchuk and Moritz, 2011). To avoid including unrealistic effects due to co-linearities or model overfitting, we do not allow



some predictors to be included if the sign of their effects are inconsistent with our understanding of western US forest fire (see Tables A1–A3).

## 4.1 Model framework

We use stepwise multiple regression to build the $P$ and $A$ models. We use multiple logistic regression to calculate our estimates of $P$ ($P_{est}$):

$$P_{est} = 1 / (1 + \exp(-\mathbf{X}_P\beta_P)), \tag{1}$$


where $\beta_P$ is a vector of logistic regression coefficients and $\mathbf{X}_P$ is a matrix of the three $S$, $C$, and $T$ composite predictors ($S_P$, $C_P$, and $T_P$, respectively), as well and their interaction terms ($S_PC_P$, $S_P,T_P$, and $C_P,T_P$), such that

$$\mathbf{X}_P\beta_P = \beta_{P0} + \beta_{P1}S_P + \beta_{P2}C_P + \beta_{P3}T_P + \beta_{P4}S_PC_P + \beta_{P5}S_PT_P + \beta_{P6}C_PT_P. \tag{2}$$


Each of the three compositive predictors, $S_P$, $C_P$, and $T_P$, is a composite of a number of $S$, $C$, and $T$ variables, where each $S$, $C$, and $T$ variable included has been selected in a stepwise process and transformed to linearize its relationship with $P$ following methods to be described below.

To model $A$, we follow a similar approach as for $P$ except that we use multiple linear, rather than logistic, regression to estimate size-weighted and normalized anomalies of $A$ ($Az_w$; details in section 4.4):

$$Az_w = \mathbf{X}_A\beta_A, \tag{3}$$

where $\beta_A$ is a vector of linear regression coefficients and $\mathbf{X}_A$ is a matrix of $S$, $C$, and $T$ composite variables ($S_A$, $C_A$, and $T_A$, respectively) and their interactions ($S_AC_A$, $S_A,T_A$, and $C_A,T_A$) such that

$$\mathbf{X}_A\beta_A = \beta_{A0} + \beta_{A1}S_A + \beta_{A2}C_A + \beta_{A3}T_A + \beta_{A4}S_AC_A + \beta_{A5}S_AT_A + \beta_{A6}C_AT_A. \tag{4}$$

In the rest of this subsection we describe the parts of the model-building framework that are common to the $P$ and $A$ models. Details specific to just the $P$ or $A$ model will be described in sections 4.2 and 4.4, respectively.





Both models are built sequentially, first constructing the spatial composite predictor ($S_x$, where subscript $x$ is $P$ for the $P$ model or $A$ for the $A$ model). Next the annual cycle composite predictor ($C_x$) and its interaction with $S_x$ ($S_xC_x$) are built. Finally the
temporal anomaly predictor ($T_x$) as and its interactions with $S_x$ and $C_x$ ($S_xT_x$ and $C_xT_x$, respectively) are built.

To construct $S_x$, we first assess the general shape and strength of the relationship between each potential $S$ predictor and the variable we are modeling, $x$, using a binned regression. We sort each potential $S$ predictor into equally sized bins (45 the $P$ model and 25 for the $Az_w$ model), and calculate the mean of $x$ for each bin. For each potential predictor we then regress the
binned mean $x$ values against the means of the binned predictor values and quantify the relationship using linear, quadratic, and cubic fits. The accuracy of each fit is assessed with the Akaike Information Criterion with a correction for low sample size (AICc) (Akaike, 1974; Hurvich and Tsai, 1989) and penalty for higher-order fits. Curve fits resulting in AICc>0 are immediately dismissed. Among the remaining curve fits, a Monte Carlo significance test is conducted in which $x$ is randomized and re-binned 100 times. Curve fits are only considered if the actual AICc is more negative than at least 95% of the AICc
values from the Monte Carlo test. Finally, the variable and curve fit combination with the most negative AICc is tentatively accepted as the initial predictor ($V_{S1}$) to represent $S_x$. Specifically, $V_{S1}$ is calculated by applying the selected curve fit to all the values of the selected variable and then $S_x$ is calculated by standardizing $V_{S1}$ relative to a mean of 0 and standard deviation of 1. An initial version of the model is then developed by applying $S_x$ as the single variable to estimate $x$. Model accuracy is assessed as correlation between modeled and observed values of $x$ (see sections 4.2 and 4.4 for details about the correlation
tests specific to the $P$ and $A$ estimates). At this point in the model-building process, the model coefficients and correlation values are recalculated 100 times when $V_{S1}$ values are randomly reordered. If the model's correlation value is not >95% of the alternative 100 correlation values, then the variable under consideration is dismissed and we consider the potential predictor that led to the next lowest AICc value in the binned regression analysis.

After $S_x$ is initially created from a single variable, we calculate residuals in $x$ and explore whether additional $S$ variables should be included within $S_x$. We do this by regressing binned means of the residuals, representing variance in $x$ not yet accounted for by the model, against the binned values for all potential $S_x$ predictors still under consideration. Notably, if the predictor variable selected in the previous step has a $\log_{10}$ counterpart, or vice versa, the counterpart is not considered in subsequent model-building steps. As before, only curve fits resulting in a negative AICc and satisfying the Monte Carlo significance test are
considered. If ≥1 curve fits satisfy these criteria, the variable and curve fit with the most negative AICc is passed on for further consideration as $V_{S2}$ by updating the calculation of $S$ by adding $V_{S2}$ to $V_{S1}$ and re-standardizing. We then re-fit the regression equation using $S$ to estimate the predictand and calculate an updated correlation between model estimates and observations. If the updated correlation is more positive than the previous correlation and is also more positive than 95% of 100 Monte-Carlo generated correlations calculated with randomized $V_{S2}$ values, then the model is updated using $V_{S1}$ and $V_{S2}$. If these correlation
criteria are not satisfied, the variable and curve fit that resulted in the next most negative AICc value is considered as a potential $V_{S2}$. This process is repeated until no additional variable and curve fit satisfies the above criteria for inclusion in $S_x$.



Next, the $C_x$ component is added, constructed in the same stepwise manner as $S_x$, where a $C$ variable is only included in $C_x$ if (1) the binned regression with residuals leads to a negative AICc that is lower than 95% of values produced when residuals are randomized 100 times and (2) model estimates of $x$ correlate more positively with observations than did the previous model's estimates and also more positively than 95% of alternative correlations calculated when the $C$ variable under consideration is scrambled randomly. A difference from construction of $S_x$ is that now the model is a multivariate regression with three predictors: $S_x$, $C_x$, and their interaction, $S_xC_x$. To avoid nonsensical interactions in $S_xC_x$ where two negative predictor anomalies, both indicating conditions inconducive to fire, would have the same effect as two positive anomalies, we positive-shift all $S_x$ and $C_x$ values by subtracting each predictor variable's minimum value before multiplying them. For the $P$ model, we subtract the lowest $S_P$ and $C_P$ values to occur among all grid-months in the calibration period. For the $Az_w$ model, we subtract the lowest $S_A$ and $C_A$ values among grid-months that cooccurred with calibration-period fire. We then calculate $S_xC_x$ as the standardized product of the positive-shifted $S_x$ and $C_x$ predictors such that the values of $S_xC_x$ have a mean of 0 and standard deviation of 1.

Finally, the same methods are used to construct $T_x$ to capture temporal variability not accounted for by $S_x$ and $C_x$. With $T_x$ included, the matrix of normalized predictors ($\mathbf{X}$) includes all 6 predictor variables shown in Equations 2 and 4 ($S_x$, $C_x$, $T_x$, and 3 interactions).

Following parameterization of the initial models, we found that some potential predictor variables not selected initially could contribute significantly if considered in a second pass. This was unsurprising because each stepwise improvement to one component of the model affects the influence of the other components through interactions. We thus perform a second pass in the model-building process in which $S$, $C$, and $T$ variables that were not selected in the original construction of $S_x$, $C_x$, and $T_x$ are given another opportunity for inclusion. In addition, we consider a small number of variables that were not considered in the first pass. For example, some variables such as temporally varying $SWE$ and fractional snow coverage do not fall cleanly into one of the three categories. Snow may be viewed as a landcover feature that inhibits fire spread or modulates the ability of climate anomalies to affect fuel moisture, in which case $S$ is appropriate, but snow presence and amount are highly variable in time. Breaking snow qualities into monthly climatologies and standardized anomalies about those climatologies is not ideal, however, as SWE and fractional coverage are highly non-normal and dominated by zeros in most places. We therefore allow, in the second round of stepwise model fitting, for monthly SWE and fractional snow coverage to be considered as both $S$ and $T$ variables. We also consider some additional $S$ variables representing distance from road as well as landcover characteristics that are not outputs from the DYNAFFOREST model. These variables are only considered in the second round because (1) we do not have a temporally varying dataset of road networks and (2) we prefer that the effect of landcover on modeled fire is dominated by variables that we can simulate with DYNAFFOREST as coupled interactors with fire. Tables A1–A3 specify the variables we only consider in the second rounds model fitting.





## 4.2 The forest-fire probability model


To model $P$, we use all available grid-months in the observed 1985–2024 forest-fire dataset to fit a logistic regression (equations 1 and 2). During this period, the 7,673 observed forest fires occurred in 7,414 unique grid-months. For more efficient model parameterization and to avoid biasing the model with conditions under which large forest fires are exceedingly improbable, we exclude grid-months from our logistic regression where mean daily $SWE$ exceeds the 99th percentile (0.86 mm)

of values coinciding with the 7,673 forest fires. Excluding the 19% of calibration-period grid-months when mean $SWE$ exceeds this value leaves a sample size of 4,321,605 grid-months with which to parameterize the $P$ model. Among these grid-months, the observed frequency of ≥1 forest fire is 0.0017.

We assess the accuracy of the logistic $P$ model using the Matthew's correlation coefficient (MCC) (Matthews, 1975), which

rewards correct positive and negative classifications and penalizes against incorrect classifications. Because $P_{est}$ is scalar (0–1), we convert $P_{est}$ to 100 potential predictions of binary fire occurrence by, for each grid-month, drawing 100 random, uniformly distributed numbers from 0–1, predicting fire occurrence (1) in all cases where the random number is less than $P_{est}$, and predicting no fire (0) when the random number is greater than $P_{est}$. This allows for calculation of 100 MCC values and we consider the mean value to represent the MCC of the model.


To construct the $S_p$ component we consider 50 potential predictors initially and 14 additional predictors in the second pass (Table A1). Variables and curve fits selected by the stepwise process to build the composite $S_p$ predictor are shown in Fig. 4a. Variables not included in the original round of model fitting but added in the subsequent round are indicated by "2nd round" in Fig. 4. The construction of $S_p$ indicates that $P_{est}$ is promoted by topographic slope, lightning frequency, high fractional forest

coverage and forest connectivity, and high prior-year precipitation total where grass and shrub cover is abundant. $P_{est}$ is reduced in areas of high housing density, near roads, in areas with high unburnable cover (barren land and water), and where the mean climatology is very wet (mean annual aridity index >2 standard deviations above the mean).

To construct $C_p$, we consider 48 potential predictors initially and 16 additional predictors in the second pass (Table A2, Fig.

4b). The annual cycle of $P_{est}$ is dominated by annual cycles in fire weather (high $HDWI$), fuel moisture (as represented by wet-day frequency, VPD, and solar radiation), and ignition potential (lightning and high $CAPE$).

To construct $T_p$, we consider 25 potential $T$ predictors initially and 12 additional predictors in the second pass (Table A3, Fig. 4c). High $P_{est}$ is promoted when $VPD$ has been anomalously high over the past 9 months and in months with high $HDWI$ and

infrequent precipitation, but $P_{est}$ can be suppressed if precipitation totals were very low between 1.5 and 0.5 years ago.





**Figure 4: Predictor variables and associated curve fits in the fire-probability ($P$) model.** Variables are in three categories: spatial ($S_p$), annual cycle ($C_p$), and temporal anomaly ($T_p$). Y-axis values in each panel indicate observed fire probabilities ($P_{obs}$; x10⁻³) not already accounted for prior to inclusion of that panel's predictor variable. Bars indicate the mean residual $P_{obs}$ values among grid-months for which the predictor variable falls within each of 45 evenly spaced bins. Red lines/curves indicate the linear, quadratic, or cubic fit used to approximate the response of $P_{obs}$ residuals to each predictor variable. With the exception of some $C_p$ predictor variables, which are scaled from 0–1, predictors are expressed as z-scores (standard-deviations





from the mean). Variable names are provided above each panel and are defined in Tables A1–A3. Panels representing variables
selected in the second round of model fitting have grey text: "*2nd round*."

The spatiotemporal distribution of $P_{est}$ generally agrees well with observations (Fig. 5). However, there is a systematic positive
bias of $P_{est}$ among very low values. In particular, among the grid-months that we excluded from model calibration due to mean
daily $SWE$ exceeding the 99$^{th}$ percentile, $P_{obs}$ was 28.71% of $P_{est}$. We therefore apply a bias adjustment to all grid-months with
$SWE$ exceeding the above threshold by multiplying $P_{est}$ in these grid-months by 0.2871. Despite the bias correction for snowy
grid-months, our model still systematically overestimates $P_{est}$ among grid months with low values of $P_{obs}$ (Fig. 5a). Among the
50% of 1985–2024 grid-months where $P_{est}$ is below the median ($2.79 \times 10^{-4}$), the mean $P_{obs}$ is 53% of modeled. We do not apply
a further correction to account for this because the positive bias among low $P_{est}$ values is of little consequence to the accuracy
of the $P$ model. The vast majority of fires are simulated to occur under higher $P_{est}$ conditions; 96% of simulated fires occur
where $P_{est}$ is above the median. Among these grid-months, $P_{est}$ scales well with $P_{obs}$ (Fig. 5a). This finding of consistently
strong model skill where $P_{est}$ is above-median generally holds true at the regional scale as well (colored dots in Fig. 5a represent
the 4 regions mapped onto Fig. 5b). Figure 5b further shows a realistic geographic distribution of mean $P_{est}$; our model captures
known areas of particularly high fire densities such as in California's Sierra Nevada and North Coast ranges, the mountainous
areas of southern Arizona and New Mexico, and a relatively remote portion of the northern Rocky Mountains in central Idaho.


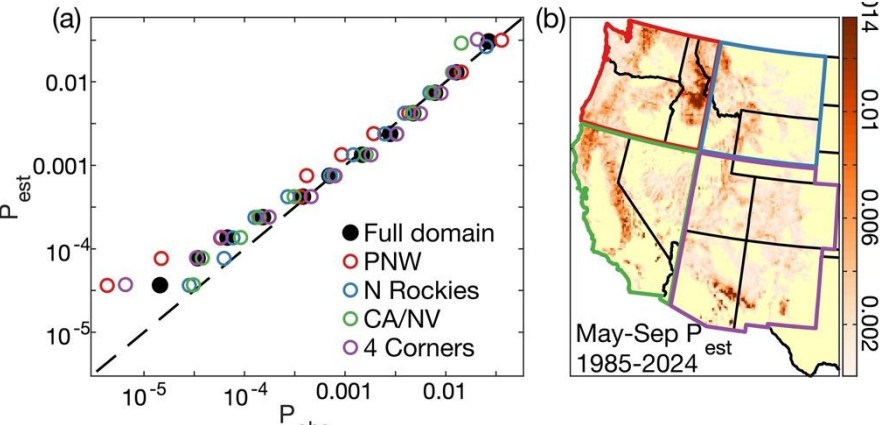

**Figure 5: Fire probability estimates.** (a) Mean observed  and estimated probabilities of grid-months with ≥1 fire ($P_{obs}$ and
$P_{est}$ , respectively) within each of 12 bins of $P_{est}$. Location of dots on the y-axis corresponds to the mean $P_{est}$ within each bin.
Filled black dots: mean observed versus simulated frequency of all grid-months in the full western US forested domain. Empty
colored dots: analysis for each of the four quadrant regions mapped in panel (b). Dashed black line: 1-to-1 line. Model estimates
shown on y-axis to aid visual interpretation of model errors. (b) Map of modeled monthly $P_{est}$ averaged over May–September
1985–2024 with boundaries of the four regions.

### 4.3 Modeling number of forest fires per month



Following Westerling et al. (2011), we use the modeled probability of ≥1 forest fire occurring in a grid-month ($P_{est}$) as a single
predictor in a logistic regression to estimate the probability that the number of fires in a given grid-month equals or exceeds
$N$, where $N$ can be 1, 2, or 3. For each possible $N$, a logistic regression is fit using $P_{est}$ from the 7,414 grid-months with ≥1
forest fire.

$$P_N = 1 / (1 + \exp(-\beta_{N0} - \beta_{N1}P_{est})), \tag{5}$$


where $N$ varies from 1–3 and the $\beta_N$ values are empirically fit logistic regression coefficients associated with each value of N.
By design, $P_N = 1$ when $N = 1$ and $P_N$ reduces as $N$ increases (Fig. 6). The maximum $N$ we consider is 3 because there are very
few occurrences of grid-months in the observed dataset with >3 fires. To prevent unrealistically large numbers of fires in a
grid-month, $P_N$ is not allowed to exceed the largest $P_N$ value that was associated with $N$ fires during model calibration.


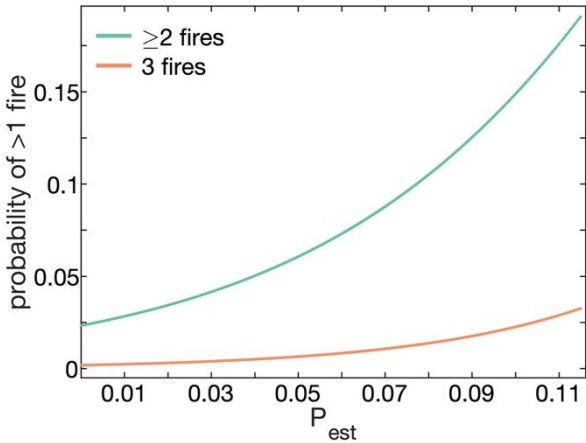

**Figure 6: Probability of more than one wildfire.** Given that ≥1 forest fire occurs in a given grid-month, the probability that
the number of forest fires equals or exceeds 2 or 3 as a function of the modeled probability of ≥1 forest fire ($P_{est}$). The maximum
number of fires in a grid-month is 3 because this is the maximum in our observational dataset.

**4.4 Area-burned modeling**

To model each fire's forested area burned ($A$), we fit a multi-variate linear regression based on spatial ($S_A$), annual cycle ($C_A$),
and temporal anomaly ($T_A$) predictor variables to estimate transformed values of $A$ for the 7,673 forest fires (eqn. 3 and 4).
Because fire sizes have a highly skewed distribution with large fires disproportionately dominating the total area burned, we
statistically transform the observed values of $A$ to quantiles and then convert the quantile values to standardized anomalies ($\sigma$)
with a normal distribution ($Az$).



Because of the disproportionate influence of large fires, we weight all regressions performed while building the $Az$ model as the logarithm of forest area burned, linearly scaled from zero to one. Weights of zero (100 ha forest area burned) were reassigned the next lowest weight. To assess accuracy of the $Az_w$ model, we use a weighted Pearson's correlation (r).


In fitting the $Az_w$ model, we initially considered 78 potential $S_A$ predictors, 58 potential $C_A$ predictors, and 47 potential $T_A$ predictors (Tables A1–A3). Because fires often burn over multiple months and may not reach large-fire (≥100 ha) status in the ignition month, the potential predictor variables for $C_A$ and $T_A$ include climate conditions in the month following the starts date of fires. In the second round we consider 22, 0, and 21 additional variables for $S_A$, $C_A$, and $T_A$, respectively. The predictor

variables and curve fits selected for the $Az_w$ model are shown in Figure 7. The variables selected for $S_A$ indicate that large fire size is promoted where forest biomass, tree height, and topographic slope are high, and where the long-term average climate is not too wet and roads and population centers are far away (Fig. 7a). Variables selected for $C_A$ indicate the annual cycle in fire size is driven by the annual cycles of fuel moisture and fire weather (Fig. 7b). Variables selected for $T_A$ indicate temporal variations in fire sizes are also dominated by fuel moisture, as represented by high $VPD$ and low $FM1000$, and high fire-

weather conditions in the month of or following ignition (Fig. 7c).

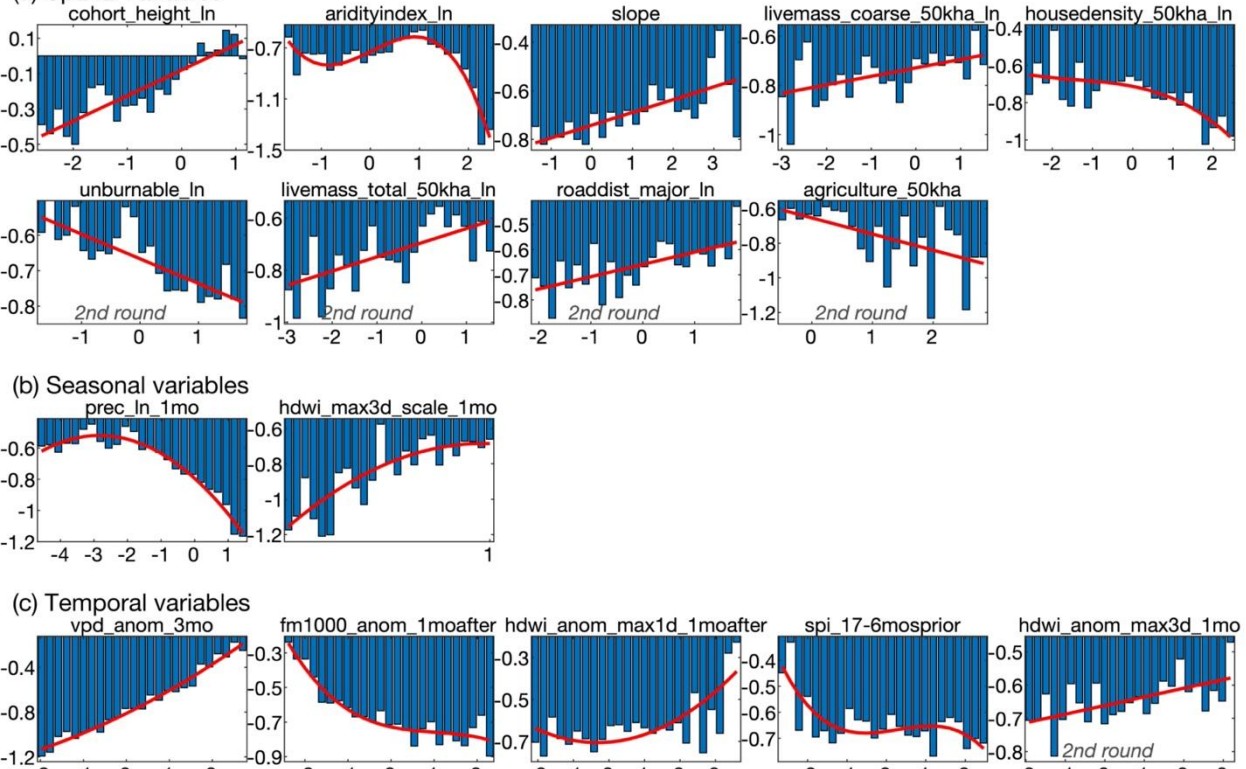



**Figure 7: Predictor variables and associated curve fits in the included in the size-weighted area burned ($Az_w$) model.** Variables are in three categories: spatial ($S$), annual cycle ($C$), and temporal anomaly ($T$). Y-axis values in each panel indicate residual $Az_w$ not already accounted for prior to inclusion of that panel's predictor variable. Most residual values are negative because the $Az_w$ predictions are inherently biased positive because the weighted regression prioritizes estimation of large fire sizes. Bars indicate the mean residual $Az_w$ among observed forest fires for which the predictor variable was split into 25 evenly spaced bins. Red lines/curves indicate the linear, quadratic, or cubic fit used to approximate the response of $Az_w$ to each predictor variable. With the exception of some $C$ predictor variables that are scaled from 0–1, units of x-axis predictors are in standard-deviations from the mean. Variable names are provided above each panel and are defined in Tables A1–A3.

### 4.4.1 Bias-correction of Az and transformation to forest area burned

Our modeled values of $Az_w$ are biased positive by an average of 0.651 σ relative to observed $Az$ ($Az_{obs}$) (Fig. 8a). This is expected because the weighted regression preferentially represents larger fires and we find no systematic tendency for the bias to vary seasonally or geographically. We apply a bias correction to calculate our model estimates of $Az$ ($Az_{est}$) as $Az_{est} = Az_w - 0.655$ σ. Although our fire-size model does not account for the majority of variability among individual $Az_{obs}$ values, it captures the underlying variability in mean $Az_{obs}$ among larger groups of fires. For each of 10 $Az_{est}$ bins, each representing an equal share of observed fires, the mean of the corresponding $Az_{obs}$ values is very near the mean of the $Az_{est}$ values (Fig. 8b). The alignment of the colored dots around the 1-to-1 line indicates that these results generally hold at the regional scale, though with tendencies to underestimate fire sizes in N Rockies and overestimate in CA/NV. Figure 8c maps the simulated distribution of the potential for large fires, highlighting California's Sierra Nevada and Coast Range and the eastern Cascades as being particularly conducive to large forest fires.

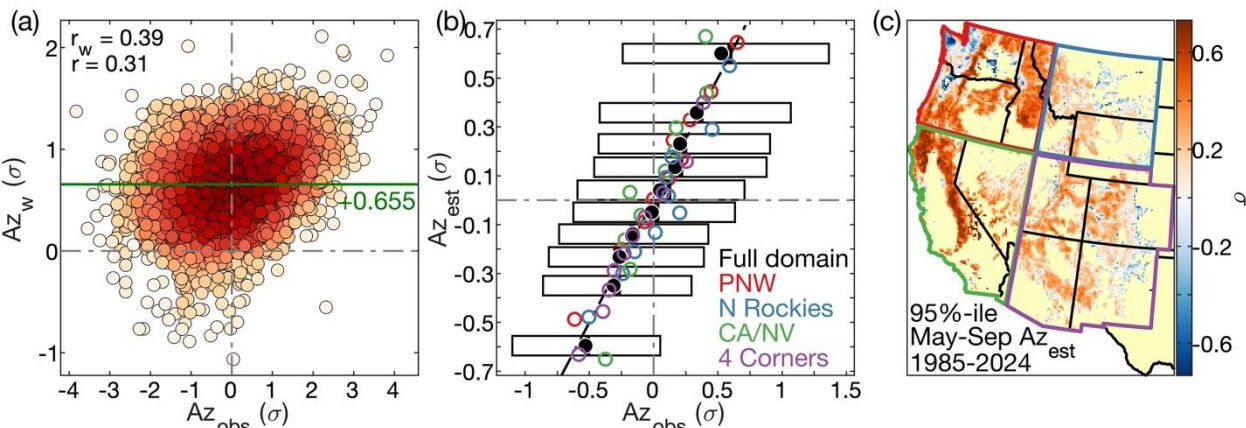

**Figure 8: Fire-size estimates.** Scatter plot of modeled, area-weighted normalized fire size anomalies ($Az_w$) versus observations ($Az_{obs}$). Redder colors indicate a higher density of sample points. The $r_w$ and $r$ values in the top-left correspond to the weighted and unweighted correlations between $Az_{obs}$ and $Az_w$. The y-axis position of the green vertical line and the green bias value correspond to the mean of $Az_w$ minus $Az_{obs}$. (b) Scatter plot of binned means of modeled $Az$ values after subtraction of the bias in $Az_w$ ($Az_{est}$) versus the means of corresponding $Az_{obs}$ values. Each black dot represents an $Az_{est}$ decile for the full western US domain, with the x- and y-axis locations representing the mean $Az_{obs}$ and $Az_{est}$ values, respectively. Horizontal extents of the corresponding boxes bound the interquartile values of $Az_{obs}$ and the vertical black line within each box is the median $Az_{obs}$.




Colored circles show binned means of $Az_{est}$ and $Az_{obs}$ when the analysis was repeated for each of the four regions (PNW: Pacific Northwest, N Rockies: Northern Rockies, CA/NV: California and Nevada, 4 Corners: the four-corner states). Black diagonal dashed line: 1-to-1 line. In (a and b), grey vertical and horizontal dashed lines cross through the zero intercepts to aid visual interpretation. Model estimates shown on y-axes to aid interpretation of model errors. (c) Map of each grid cell's 95th percentile of May–Sep $Az_{est}$ during 1985–2024 to show geographic variability in the potential for an existing fire to grow very large.

### 4.5 Accounting for stochastic variability

Across the western US and within the four regional quadrants, interannual variations in modeled $P$ and mean $Az$ generally correlate well with observations, but simulated interannual variability is systematically muted relative to observations (Fig. 9). This is expected, as the occurrences and sizes of individual fires are highly stochastic. For more realistic representation of variability in our simulations of fire occurrences and sizes, we add semi-random variability to each modeled value of $P_{est}$ and $Az_{est}$. The distributions of random variations are constrained empirically by the distributions of errors in $P_{est}$ and $Az_{est}$.

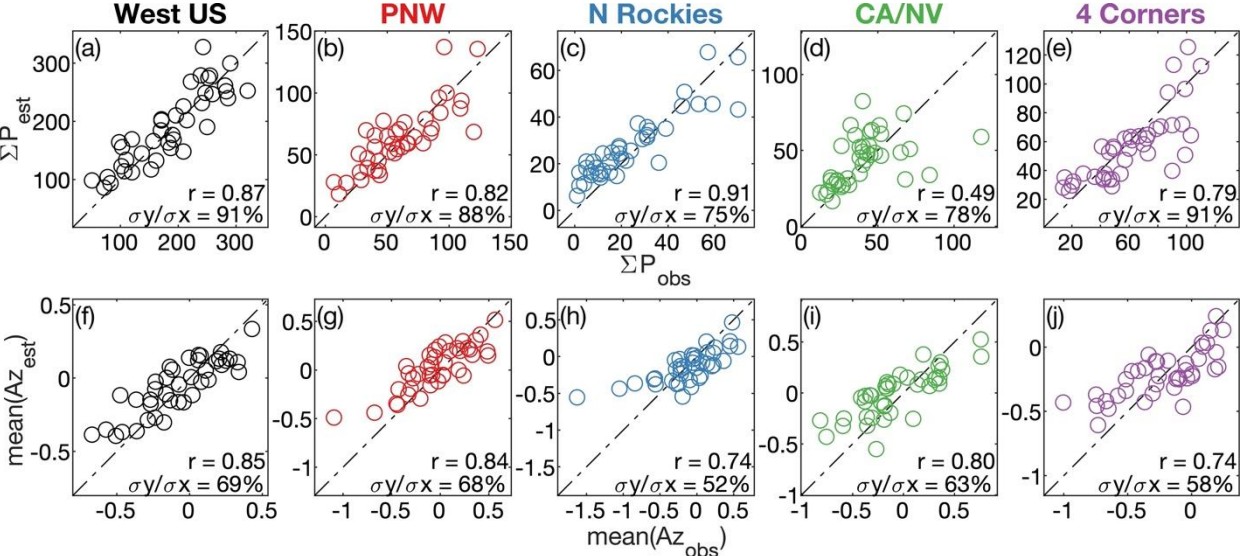

**Figure 9: Interannual variability.** Scatter plots of annual modeled versus observed forest-fire probability ($P$) and normalized fire-size anomalies ($Az$) for the western US and each of the four quadrant regions, 1985–2024. (a–e) Modeled annual sum of $P$ across all grid-months ($\Sigma P_{est}$) versus observed annual sum of grid-months with ≥1 forest fire ($\Sigma P_{obs}$). (f–j) Modeled annual mean of $Az$ ($Az_{est}$) corresponding to the grid-months of the observed fires versus observed annual mean of $Az$ ($Az_{obs}$). Diagonal dashed lines are 1-to-1 lines. Model estimates shown on y-axes to aid interpretation of model error. Correlation (r) indicates Pearson's correlation between observed and modeled time series. The σy/σx values express the standard deviation of the modeled time series as a percentage of the standard deviation of the observed time series.

#### 4.5.1 Stochastic variations in fire probability

The distribution of uncertainty around any value of $P_{est}$ is difficult to characterize because fire probability in a given grid-month can only be observed as binary, and errors in $P_{est}$ can only be assessed by comparing mean values of $P_{est}$ to $P_{obs}$ across



many grid-months. However, quantification of error in $P_{est}$ averaged across many grid-months does not provide direct guidance
as to the distribution of errors surrounding any single grid-month's $P_{est}$ value. In exploratory analysis we found that the
distribution of $P_{est}$ uncertainty does not scale predictably as a function of $P_{est}$ (e.g., errors are not systematically larger for larger
$P_{est}$ values) so we include stochasticity in our modeling of $P$ by simply adjusting $P_{est}$ with observed sequences of regionally
averaged errors.

To identify regions where temporal variability in $P_{est}$ is relatively coherent, we perform a rotated principal components analysis
(PCA) on monthly regional errors. Initially, we divide our western US forested study domain into 63 regions based on the map
of coterminous US pyromes from Short et al. (2020). To reduce the number of regions, we merge each of the 61 pyromes with
that averaged fewer than seven fires/year during 1985–2024 with the nearest pyrome, producing 10 forested regions with
adequate fire frequencies for characterization of monthly error in $P_{est}$. For each region we calculate monthly sums of $P_{est}$ and
$P_{obs}$, calculate 3-month running means centered on the middle month ($P_{est3}$ and $P_{obs3}$) to reduce the effects of extreme $P_{obs}$
outliers, and define the monthly error ($P_{error}$) as $P_{obs3}/P_{est3}$. To identify groups of regions with relatively coherent variability in
$P_{error}$, we perform a PCA on the 10 time series of $P_{error}$, and retain the six principal components (PCs) with eigenvalues $\geq1$ as
distinct modes of variability. The loadings associated with these PCs are rotated using the varimax method and multiplied
against $P_{error}$ to reproject the original $P_{error}$ variance onto new rotated PC time series. The 10 original pyrome groups are
combined into six new groups of relatively coherent $P_{error}$ variability based on correlation between $P_{error}$ and PCr($P_{error}$) (Fig.
10).

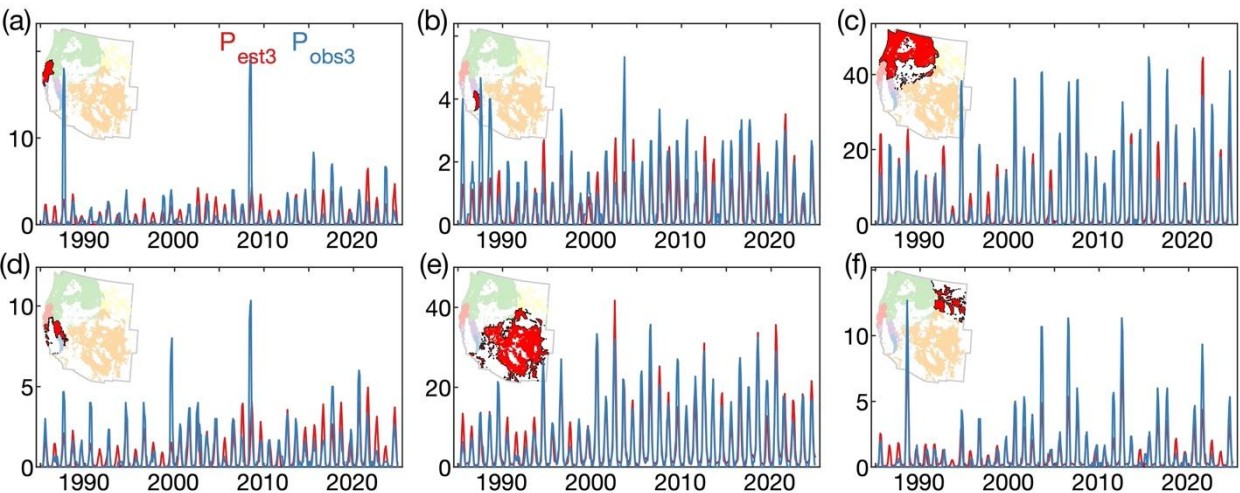

**Figure 10: Intra-annual error in modeled fire probability by pyrome group.** Time series of 3-month running means of the
modeled (red; $P_{est3}$) and observed (blue; $P_{obs3}$) monthly sums of grid cells with $\geq1$ forest fire in each of 6 pyrome groups. Each
group is composed of a group of pyromes (Short et al., 2020) with similar time series of monthly error in modeled fire
probability ($P_{error} = P_{obs3}/P_{est3}$). In each panel, the bright red area in the map indicates the pyrome group represented by the
time series and the other 9 groups are infilled with lighter colors.





To include stochastic variability in our model simulations, we calculate an adjusted version of *Pest* ($P_{estadj}$) by multiplying each simulated calendar year of $P_{est}$ values by a randomly drawn year of $P_{error}$ from the 40-year model calibration period, where each month' map of $P_{error}$ represents the regions shown in Fig. 10 (to avoid extreme values we bound $P_{error}$ between 0.33 and 3). This approach preserves realistic $P_{error}$ autocorrelation both spatially and between months. To demonstrate the effectiveness of this approach at eliminating the bias toward too little temporal variability in $P_{est}$ (shown previously in Fig. 9a–e), we produce

a 1,000-member ensemble of $P_{estadj}$ (Fig. 11). Including errors in our simulation of $P$ successfully gives $P_{estadj}$ (middle box plots in Fig. 11) a wider distribution than $P_{est}$ (left box plots) that is generally better aligned with observations (right box plots). The percentage value above each set of box plots in Fig. 11 indicates how the median standard deviation of annual simulated sums of $P_{estadj}$ compares to the observed standard deviation. These values are no longer systematically below 100% (compare to percentage values in Fig. 9a–e), indicating that our approach improves the realism of temporal variability in simulated $P$.


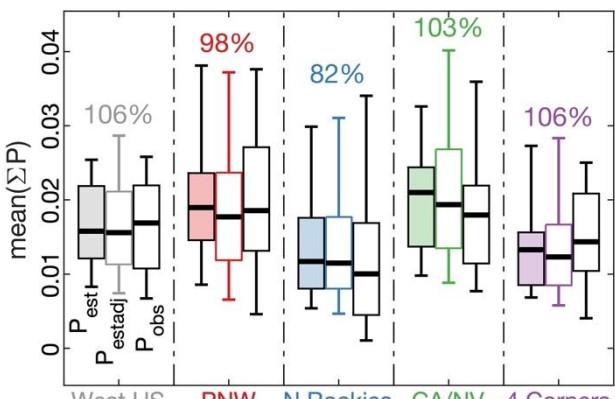

**Figure 11. Distributions of modeled and observed interannual fire probability.** Box plots of annual observed and modeled annual sums of the probability of ≥1 fire/month ($P$) averaged across all forested grid cells (mean($\Sigma P$)) in the West US study region and the four quadrant regions. For each region, the light-colored boxplot on the left represents the distribution of the

originally modeled time series of $P$ ($P_{est}$): thick line is median annual $\Sigma P$ value, box bounds interquartiles, whiskers bound inner 90% range. The boxplot in the middle represents the mean distribution across 1,000 simulated time series of $P_{est}$ after the adjustments to include random errors ($P_{estadj}$). The white box plot on the right represents the distribution of observed sums of mean $P$ ($P_{obs}$). Percentage numbers indicate the magnitude of the mean standard deviation of the 1,000 simulated time series of annual $\Sigma P_{estadj}$ relative to the standard deviation of the observed time series. Differences between these values and the

percentages provided in Fig. 9a–e are due to the inclusion of error in the 1,000 simulations represented here. Values of annual $\Sigma P$ are averaged across all grid cells for each region to reduce the influence of large regional differences in $\Sigma P$ in the figure.

**4.5.2 Stochastic variations in fire size**

The distribution of uncertainty around estimates of $Az_{est}$ is easier to assess than that of $P_{est}$ because error in $Az_{est}$ ($\varepsilon Az_{est}$) can be quantified for each fire. In addition, errors in $Az_{est}$ are normally distributed and increase as a function of $Az_{est}$ (Fig. 8b). As $Az_{est}$

increases, the spread among corresponding $Az_{obs}$ values widens but remains symmetrical. When we bin $Az_{est}$ into deciles, the



standard deviation of $\varepsilon Az_{est}$ scales linearly with $Az_{est}$ (Fig. 12). The relationship detected at the large scale of the western US also remains generally consistent at the regional scale, though $\varepsilon Az_{est}$ is consistently higher than the west-wide mean in CA/NV and consistently lower in N Rockies. Overall, we conclude that we can characterize the uncertainty $Az_{est}$ with reasonable accuracy by simply treating it as a linear function of $Az_{est}$ itself, though future work should diagnose and ideally resolve regional

variations in mean $\varepsilon Az_{est}$.

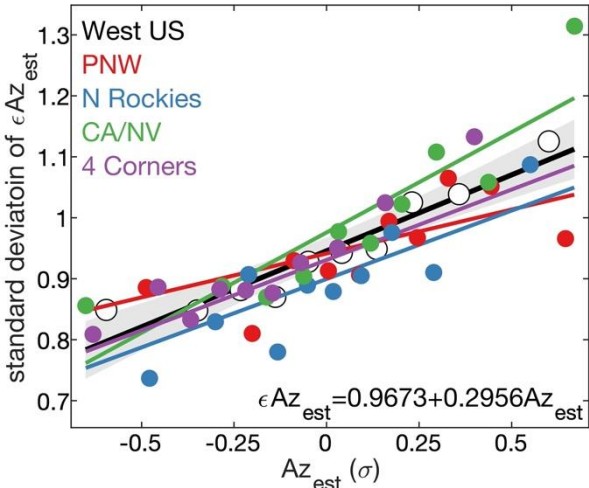

**Figure 12: Variability among modeled fire-size errors.** Standard deviation of error in estimates of normalized fire-size anomalies ($\varepsilon Az_{est}$) as a function of $Az_{est}$ for (white dots, black regression line) the entire western US forested domain and the

four quadrant regions within: (red) Pacific Northwest, (blue) Northern Rockies, (green) California and Nevada, and (purple) Four Corners. $\varepsilon Az_{est}$ is the observed normalized fires-size anomaly ($Az_{obs}$) minus $Az_{est}$. For each domain, $Az_{est}$ values associated with observed fires are binned into deciles and, for each decile, standard deviation of $\varepsilon Az_{est}$ is plotted against mean $Az_{est}$. Regression lines show the least-squares fit for each domain and the grey area bounds the 95% confidence interval around the black regression line for the full West US domain, which corresponds to the equation at the bottom of plot.


For each simulated value of $Az_{est}$ we calculate an adjusted $Az$ estimate ($Az_{estadj}$) by adding an error value drawn from a normal distribution with a mean of zero and a standard deviation of $\varepsilon Az_{est}$, where $\varepsilon Az_{est}$ is calculated as a linear function of $Az_{est}$ following the equation in Fig. 12. Based on a 1,000-member ensemble of simulated $Az_{estadj}$, this method of widening the distribution of $Az_{est}$ aligns the distribution of $Az_{estadj}$ with observations (Fig. 13).




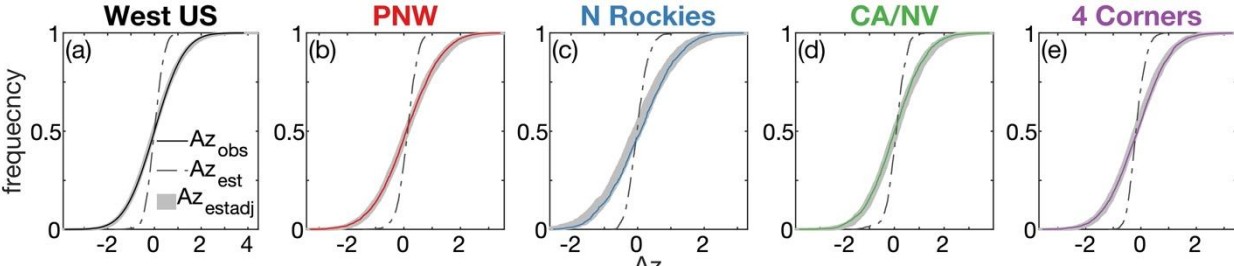

**Figure 13: Effect of adding errors on the distribution of modeled fire sizes.** Cumulative distribution functions of observed and modeled normalized fire-size anomalies ($Az$) for (a) the whole western US domain and (b–e) the quadrant regions. Thin solid lines represent observed $Az$ ($Az_{obs}$). Dashed lines represent simulated $Az$ before including error ($Az_{est}$). Grey areas represent 1,000 simulations of $Az$ after adjustment to include errors ($Az_{estadj}$).

Adding error to $Az_{estadj}$ enhances the interannual variability of mean $Az_{estadj}$ (Fig. 14). However, there is still a tendency toward too-little variation in $Az_{estadj}$. This is likely because errors in our estimates of $Az$ ($\varepsilon Az_{est}$) are spatially and temporally autocorrelated. We do not account for this because imposing realistic spatiotemporal covariance among $\varepsilon Az_{est}$ values would risk overfitting the model and reducing its interpretability.

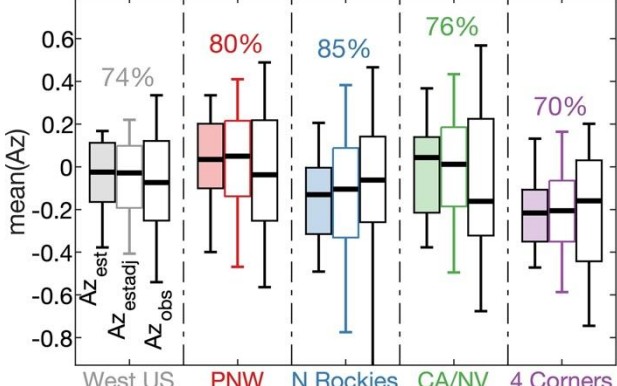

**Figure 14: Distributions of modeled and observed interannual variability in mean standardized fire size.** Box plots of annual modeled and observed annual means of normalized fire-size anomalies ($Az$) in the western US (grey) and the four quadrant regions (colors). For each region, the light-colored boxplot on the left represents the distribution of the originally modeled time series of $Az$ ($Az_{est}$). The middle boxplot represents the average distribution among 1,000 simulated time series of $Az_{est}$ after the adjustments to include random errors ($Az_{estadj}$). The white box plots on the right represent the annual time series of observed $Az$ ($Az_{obs}$). Boxes bound inter quartiles, whiskers bound 5th and 95th percentiles, and thick black bars represent medians of annual values. Percentages indicate the magnitude of the mean standard deviation among the 1,000 simulated time series of mean($Az_{estadj}$) relative to the standard deviation of the time series of mean ($Az_{obs}$). Differences between these values and the percentages in Fig. 9f–j are due to inclusion of error in the 1,000 simulations represented here.

## 4.6 Transformation of normalized fire-size anomalies to area burned





Previous work has shown that fire sizes can be effectively approximated by a positively-skewed generalized pareto (GP) distribution (Buch et al., 2023; Preisler et al., 2011; Westerling et al., 2011). We transform all values of $Az_{estadj}$ to hectares of

forest area burned ($Agp_{est}$) by assuming that fire sizes follow a GP distribution with the shape and scale parameters estimated from the observed forest fire sizes. However, a comparison of the distribution of observed $A$ ($A_{obs}$) versus the GP-transformed values calculated by back-transforming $Az_{obs}$ using the empirical GP distribution parameters ($Agp_{obs}$) reveals a bias in the $Agp_{obs}$ distribution because the GP is an imperfect representation of the true distribution of $A_{obs}$ (Fig. 15a). We quantify the observed bias ($Agp\_bias\_log10$) as $\log_{10}(Agp_{obs})$ minus $\log_{10}(A_{obs})$, which we plot as a function of $\log_{10}(A_{obs})$ in Fig. 15b. Much

of the bias arises because the $A_{obs}$ distribution has a lower bound of 100 ha (Fig. 15a), which causes the most frequent, small values of $Agp_{obs}$ to be too small and the least frequent, largest values of $Agp_{obs}$ to be too large.

To reduce shortcomings of the GP distribution we bias correct such that the bias-corrected observed fire sizes ($Abc_{obs}$) take on the distribution more consistent with that of $A_{obs}$ (Fig. 15c). This is done by estimating $Agp\_bias\_log10$ ($Agp\_bias\_log10\_est$)

as a 4$^{th}$-order function of $\log_{10}(Agp_{obs})$ for small fires ($Agp_{obs} < 225$ ha) and as a 5$^{th}$-order function of $\log_{10}(Agp_{obs})$ for larger fires (Fig. 15b). Specifically, $Abc_{obs}$ is calculated by subtracting $Agp\_bias\_log10$ from $\log_{10}(Agp_{obs})$ and transforming the $\log_{10}$ values back to normal, thereby restoring $Abc_{obs}$ to nearly the original distribution of $A_{obs}$. In simulations, bias-corrected fire sizes ($Abc_{est}$) are calculated in the same way except $Agp\_bias\_log10\_est$ is calculated as a function of $\log_{10}(Agp_{est})$ rather than $\log_{10}(Agp_{obs})$. The grey shading behind the blue and red points in Fig. 15c represents an ensemble of 1,000 simulations of

$Abc_{est}$, where in each simulation we estimate the 7,673 values of $A_{obs}$. The strong overlap between the grey, blue, and red CDFs in Fig. 15c indicates that our method is effective at producing realistic fire-size distributions. To prevent unrealistically large bias estimates in simulations, values of $Agp\_bias\_log10\_est$ should not be allowed to exceed the empirically calculated range of $Agp\_bias\_log10$ values.

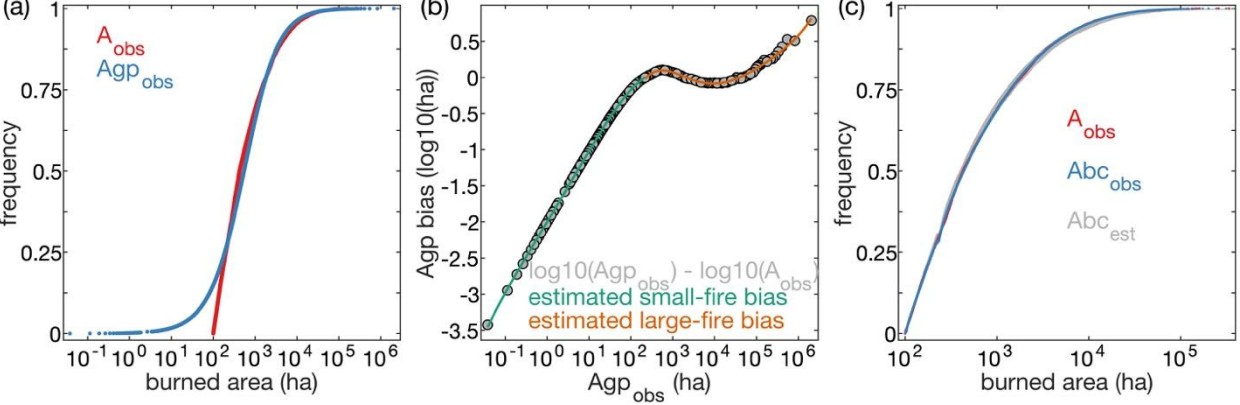


**Figure 15. Bias correction of fire-size distributions.** (a) Cumulative distribution function (CDF) of the observed fire sizes (red, $A_{obs}$) and the same observed fire sizes after being turned into quantiles and then back-transformed to hectares (ha) based on the observed generalized pareto (GP) distribution parameters (blue, $Agp_{obs}$). (b) Scatter plot of the bias in $Agp_{obs}$ caused by





the imperfect match between the actual fire-size distribution and that estimated by the GP. For small fires <225 ha, the green
curve represents a 4th-order curve of the $Agp$ bias as a function of $A_{obs}$. For larger fires, the orange curve represents a 5th-order
fit. (c) Comparison of the CDFs of (red) observed fire sizes ($A_{obs}$; same as in panel (a)) and (blue) bias-corrected observed fire
sizes ($Abc_{obs}$), where, $A_{obs}$ values were first converted to normalized fire size anomalies ($Az_{obs}$), then back transformed to
hectares assuming a generalized pareto distribution ($Agp_{obs}$), and finally bias corrected based on the curve fits in (b). The CDFs
of $A_{obs}$ and $Abc_{obs}$ are overlaid on the range of CDFs produced from 1,000 simulations of modeled fire sizes (grey, $Abc_{est}$),
where, in each simulation, the model is used to estimate the observed fire sizes.

### 4.7 Cross-validation

To assure the skill of WULFFSS is not due to overfitting, we perform temporal and spatial cross-validations. In the temporal
cross validation, we retrain the models 13 times, each time withholding a period of 3–4 consecutive years such that each year
in the 1985–2024 calibration period is withheld once from the training period. We then use each of the 13 models to simulate
fire for the withheld period. For the spatial cross-validation we again produce 13 models, now withholding from each model
calibration a contiguous region approximately 500 x 500 km in area. Each model is then used to simulate 1985–2024 fire for
its withheld region. For each cross-validation approach, a full set of out-of-sample simulation outputs are produced for the
western US for 1985–2024 and correlated against observations for assessment of out-of-sample skill.

## 5 Model Performance

The WULFFSS simulations of frequency and extent of western US forest fires are generally highly skilled (Figs. 16 and 17).
The mean of a 100-member ensemble of model simulations accounts for 72% (r=0.85) of the observed interannual variability
in western US forest-fire frequency (Fig. 16a, left side). Model performance remains high out-of-sample. In the 13-fold
temporal cross-validation, the cross-validated correlation between observed and ensemble-mean simulated annual fire
frequency remains high at 0.82. The model performs similarly well (r=0.83) in the 13-fold spatial cross-validation. WULFFSS
also accurately simulates the mean annual cycle of forest-fire frequency and correlation between the full monthly time series
of observed and modeled fire frequency is strong (r≥0.91) (Fig. 16a, right side).

The model generally performs well at the regional level, accounting for ≥62% of variability in annual fire frequency in PNW
(r≥0.79; Fig. 16b), ≥77% in N Rockies (r≥0.88; Fig. 16c), and ≥56% in 4 Corners (r≥0.75; Fig. 16e). The CA/NV region is an
exception to the strong model performance (Fig. 16d), where the ensemble-mean accounts for just 17–18% of interannual fire-
frequency variability (r=41–0.42), due mostly to large underestimates in 1987 and 2008 as well as recent overestimates in
2020–2022 and 2024. Reasons for model underperformance in CA/NV are likely numerous but may include increased
resources for fire detection and suppression in California, increased public awareness of fire hazards, post-2011 reductions in
fuel continuity due to drought and related bark-beetle outbreaks that our forest-ecosystem modeling does not capture, and
inadequate representation of lightning. The model also majorly underestimates 2024 fire frequency in PNW, due to a failure
to capture the large number of fires in Oregon and southwest Idaho that ignited from outbreaks of dry lightning in mid and late





July. While WULFFSS does consider long-term mean patterns of lighting activity, it does not model fire as a function of temporal variability in lightning because the only long-term lighting dataset we are aware of (from the NLDN) has temporal instabilities due to instrumental changes and it does not cover the full model-calibration period. While *CAPE* is considered as

a temporally varying model due to its coincidence with lightning, high *CAPE* is also associated with precipitation, limiting its value as a proxy for dry lightning.

The model does generally well at capturing regional differences in the mean annual cycle of fire frequency (Fig. 16, right-hand panels). For example, the model correctly simulates that peak monthly fire frequency occurs in August in PNW and N Rockies

but in June–July in 4 Corners (Fig. 16b–e). WULFFSS accurately simulates regional differences in the timing of onset and termination of the mean annual fire starts. On the other hand, the spatial cross-validation reveals that when training data are withheld from 4 Corners, the model underestimates fire frequencies throughout the fire season in that region (Fig. 16e).





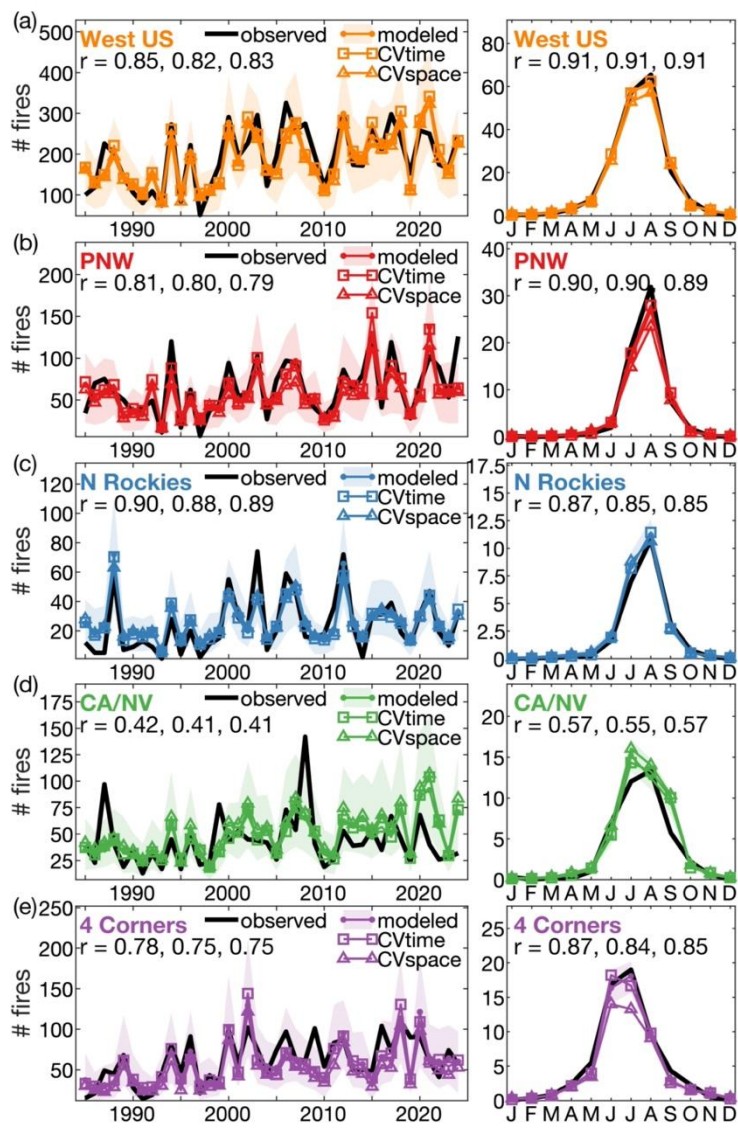

**Figure 16: Modeled versus observed forest-fire frequency.** Plats for (a) the western US and each of the four quadrant regions: (b) Pacific Northwest (PNW), (c) Northern Rockies (N Rockies), (d) California and Nevada (CA/NV), and (e) 4 Corners. Panels on the left show annual frequency of (black) observed and (colored) modeled forest fire. Panels on left show annual values and panels on right show the mean annual cycle of monthly values. The three colored lines indicate 100-member ensemble means of (dots) the fully calibrated model, (squares) the 13-fold temporally cross-validated models (CVtime), and (triangles) the 13-fold spatially cross-validated models (CVspace). Colored shading bounds the inner 95% of ensemble members of the fully-calibrated model. In each panel, the three correlation values (r) indicate Pearson's correlation between (r) observations and the ensemble means from the fully calibrated model, the 13-fold temporally cross-validated models, and the 13-fold spatially cross-validated models, respectively. In the annual-cycle panels on the right, correlation values indicate correlation between the full observed and modeled time series of monthly fire-frequency over 1985–2024, not the mean annual cycle.





Model performance is also strong in terms of area burned, accounting for 83% (r = 0.91) of interannual variability in the logarithm of area burned when fully calibrated and ≥76% (r≥0.87) in our cross-validated exercises (Fig. 17a). At the regional scale, model performance remains strong, accounting for 66–79% of cross-validated variability in PNW, N Rockies, and 4

Corners, and ≥56% in CA/NV. The model also reproduces observed regional differences in relatively nuanced time-series characteristics of annual area burned. For example, the model captures the tendency for interannual burned-area variability to be dominated by extreme years in N Rockies and 4 Corners, and for variability to be more evenly distributed in PNW and CA/NV (Fig. 17b–e). The model also generally captures the mean annual cycles and sub-annual variations in area burned, though in CA/NV our model consistently over-estimates burned areas throughout the fire season (Fig. 17d). This is partly due

to overestimating July and September fire frequencies (Fig. 16d), but also due to a systematic overestimation of fire sizes year-round in that regions. In 2020, however, WULFFSS grossly underestimates CA/NV area burned, and to a lesser extent in PNW (Fig. 17b,d). Potential contributing factors include rare lightning storms from tropical storm Fausto in August 2020, two extreme heat waves in the days to weeks immediately following the lightning storms, and overstretched suppression resources due to a high concentration of large forest fires in California and Oregon and the COVID-19 pandemic.




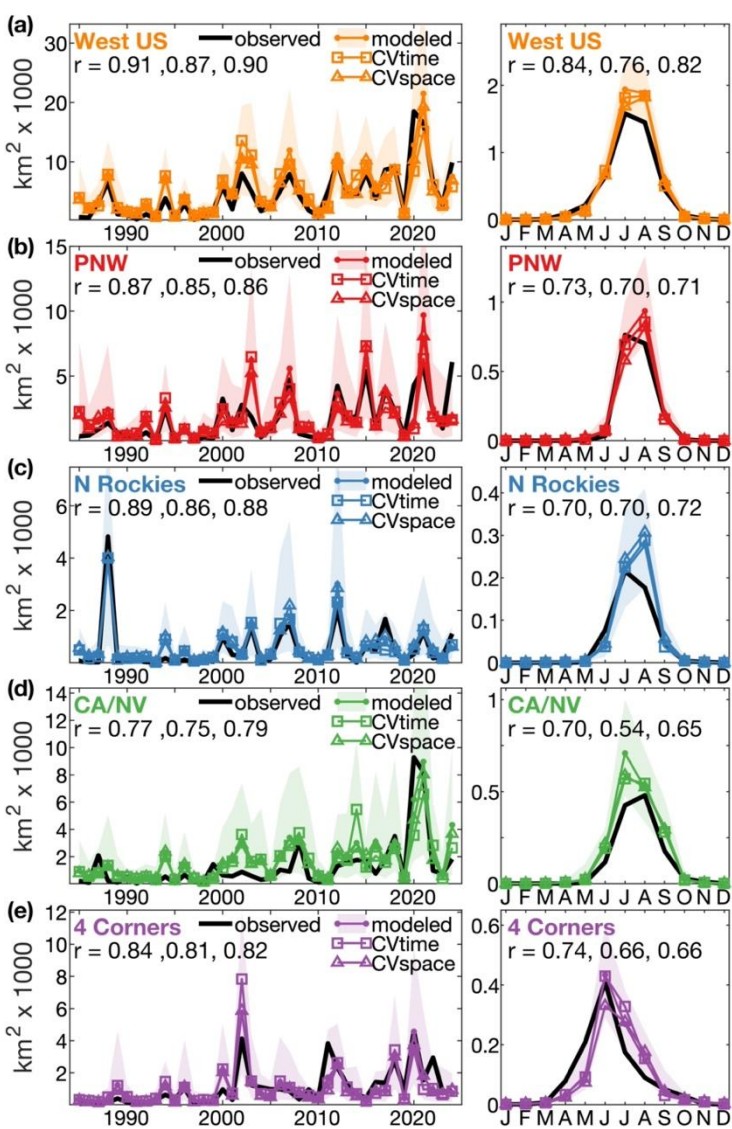

**Figure 17: Modeled versus observed forest-fire area.** Plots for (a) the western US and each of the four quadrant regions: (b) Pacific Northwest (PNW), (c) Northern Rockies (N Rockies), (d) California and Nevada (CA/NV), and (e) 4 Corners. Panels on the left show annual (black) observed and (colored) modeled forest area burned. Panels on left show annual values and panels on right show the mean annual cycle of monthly values. The three colored lines indicate 100-member ensemble means of (dots) the fully calibrated model, (squares) the 13-fold temporally cross-validated models (CVtime), and (triangles) the 13-fold spatially cross-validated models (CVspace). Colored shading bounds the inner 95% of ensemble members of the fully-calibrated model. In each panel, the three correlation values (r) indicate Pearson's correlation between (r) observations and the ensemble means from the fully calibrated model, the 13-fold temporally cross-validated models, and the 13-fold spatially cross-validated models, respectively. In the annual-cycle panels on the right, correlation values are for the full observed versus modeled time series of monthly forest area burned over 1985–2024, not the mean annual cycle.

## 6 Discussion, strengths, and limitations



The WULFFSS simulates the monthly gridded probabilities and sizes of forest fires in the western US as a function of land cover, topography, human population, and climate. The model uses standard regression-based statistical methods, which constrains flexibility but enhances interpretability and reproducibility. We suggest the skill of our model should serve as a benchmark for more complex but methodologically opaque modeling efforts.

Our model has high skill. It simulates realistic characteristics of fire such as annual cycles, ranges of interannual variability, and fire-size distributions, as well as inter-regional differences in these characteristics. The model also has strong out-of-sample skill when reconstructing observed variations in forest-fire activity for time periods or regions withheld from the training data. This suggests that the model can reliably simulate western US forest-fire activity under idealized historical or projected conditions as long as those conditions are not far beyond those that occurred during the model training period.

The model can be easily updated as additional or improved records of observed wildfires become available. Updates and improvements of the observed fire record are enabled by our streamlined method to easily update our WUMI2024a database with newly available wildfire data (Williams et al., 2025). Our model's ability to produce trustworthy simulations under future, warmer climate scenarios will likely improve over time as more climate extremes and their effects on forest fires are observed.

A unique feature of WULFFSS is that it was developed in parallel with the forest-ecosystem model, DYNAFFOREST, specifically to enable coupled simulations in which fire and forest ecosystems interact. This is important for several reasons. First, we are motivated to simulate and understand more features of fire beyond event frequency and area burned. By coupling with an ecosystem model, we can also simulate fire severity, biomass consumed, and ecosystem transitions, all crucial for anticipating changes to ecosystem health, pollution, hydrology, or terrestrial carbon storage. Further, as vegetation responds to changes in climate and fire behavior, these responses will feed back to modulate fire-climate relations. Coupling between fire and forest-ecosystem models is therefore essential for plausible projections of western US forest fire activity beyond the next couple decades.

Another feature of WULFFSS is its computational efficiency, which allows for large ensembles of simulations. A standard laptop can simulate several decades of forest fire across the western US in seconds, enabling easy generation of hundreds or thousands of simulations. This is important under climate warming because forest-fire sizes appear to respond exponentially to positive forcings such as warming and drying, which should cause the range of internal variability of area burned to grow under continued warming in many forested regions of the western US. Indeed, the range of modeled uncertainty in total forest-fire area is much wider in high-*VPD* years (Fig. 18). Although running WULFFSS while coupled within the DYNAFFOREST model is considerably more computationally expensive, DYNAFFOREST was also designed to facilitate large simulation ensembles and it is feasible to run tens of century-scale coupled simulations in the matter of days on a high-performance computer cluster. With a large ensemble of tens of historical or future coupled forest and fire simulations, one can explore the





mean response (e.g., aboveground biomass consumed) to a given forcing as well as the uncertainty around the mean. Further, in an ensemble of coupled simulations where each represents a plausible realization of fire effects on forest biomass, connectivity, etc., then these forest outputs can be used to force the WULFFSS in uncoupled mode to greatly enhance the ensemble size in terms of simulated fire frequencies and burned areas.

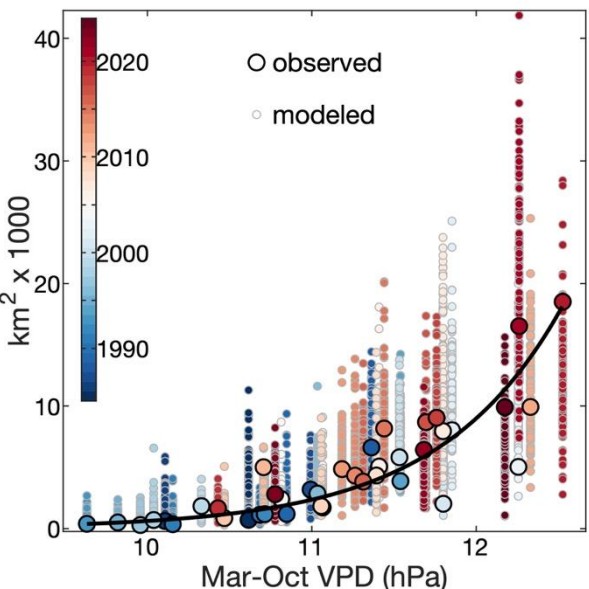

**Figure 18: Western US annual forest-fire area versus March–October vapor-pressure deficit (*VPD*).** Large dots with black outlines are observations and the black curve is the least-squares regression line relating the logarithm of observed area burned to observed *VPD*. Small dots with grey outlines are outputs from an ensemble of 100 simulations under identical forcings, including observed climate (ensemble spread due to stochastic errors added to modeled estimates of fire probabilities and sizes). Colors correspond to years from 1985–2024.

There are a number of caveats, some of which represent opportunities for improvement while others are structural features of our approach. Opportunities include consideration of changing road networks in the past, use of road networks to more intelligently map the distance of forested areas to human population, and addition of aboveground utility lines and their ages. In addition, the model's ability to capture the effects of spatiotemporal changes in fuel characteristics is limited by a lack of spatially continuous observational data covering the four-decade model-calibration period. For example, while the model does account for the majority of the observed increase in western US annual forest-fire area since 1985, it systematically overestimates burned area in the first half of the record. One likely explanation is that the DYNAFFOREST datasets we use to parameterize WULFFSS do not fully represent fire-promoting trends in fuel amount, connectivity, and structure in recent decades. Because DYNAFFOREST is a single-cohort model, it does not explicitly simulate understory fuels, so variables related to vertical forest structure and ladder fuels are not currently considered by WULFFSS. As spatially continuous remotely





sensed fuels datasets, which are so far only available for smaller regions (Hudak et al., 2020), become available across the
western US, this will almost certainly improve our ability to simulate historical probability and size.

Another limitation is that WULFFSS and DYNAFFOREST do not explicitly represent non-forest vegetation.
DYNAFFOREST assumes non-forest is grass, but does not explicitly simulate grass and shrub growth and decomposition.
Representation non-forest fuel dynamics would likely improve our ability to simulate fire events, particularly near the dry
edges of forests and when and where simulated forest biomass is relatively sparse.

Likewise, more mechanistic consideration of fuel-moisture dynamics would improve the realism of WULFFSS. In the current
parameterization we mechanistically model snowpack and allow this to affect our calculation of dead fuel moisture, but the
NFDRS formulations we use to estimate dead fuel moisture are relatively simple and non-mechanistic. Live fuel moisture
would likely improve model skill beyond the skill yielded from our estimates of dead moisture (Rao et al., 2023), but our
current approach instead relyies on climate predictors to implicitly represent live fuel moisture. More thoroughly representing
the complexity of moisture dynamics in an internally consistent framework that can be coupled with our ecosystem simulations
would likely enhance the skill of WULFFSS. That said, fuel-moisture simulations are challenging due to the limited availability
ground-truth measurements across the complexity of fuel moisture dynamics related to species, fuel sizes, types, ages, soil
type and geology, rooting depth, position within the vertical profile of the forest, stand density, and live versus dead status.

Another opportunity for improvement is to explicitly simulate fire spread. Currently, WULFFSS only estimates the final forest
area burned by each simulated fire. When coupled within DYNAFFOREST, the ignition of a given simulated fire is assigned
to a random 1-km forested-grid cell within the 12-km grid cell of WULFFSS and the fire spirals through adjoining or nearby
forested areas until the pre-determined fire size is achieved or no nearby forested grid cells remain. Future improvements to
WULFFSS should model fire spread (e.g., probabilistic determinations of sub-grid ignition location, sub-month ignition date,
fire-spread duration, and daily wind velocity) while maintaining computational efficiency. Related to processes affecting fire
spread, with the exception of our consideration of CAPE to represent likelihood for lightning or plume developing, we currently
only rely on surface climate to represent potential for rapid fire spread. Future work should consider how fire spread is linked
to three-dimensional atmospheric dynamics.

A limitation to essentially all fire models that can operate across large geographies, especially statistical models like
WULFFSS, is that the observations used for model parameterization inherently reflect the impacts of modern human society.
These impacts include non-lightning ignitions and restricted fire sizes due to suppression, as well as the indirect effects of
humans on fuels (e.g., fuel accumulation due to fire suppression) and climate. Future improvements should include
distinguishing human- versus lightning-caused ignitions. More challenging is to estimate fire sizes in the absence of
suppression or under changes to suppression practices. The North American Fire-Scar Network, a database of historical fire





scars in trees (Margolis et al., 2022), could provide guidance as to how simulated fire sizes could be adjusted to represent a fire regime with little or no suppression.


However, spatiotemporal differences in human behavior cause uncertainty in WULFFSS, even in the observed period. In 2020, for example, the observed area burned in the western US was on the upper fringe of values simulated by WULFFSS (Fig. 17). Interestingly, WULFFSS accurately simulates fire frequency in 2020, but systematically underestimates 2020 fire sizes. A likely explanation is that, when a rare summertime lightning event coincided with hot and dry conditions to produce widespread

wildfire activity, coupled with the COVID-19 pandemic, suppression efforts had difficulty keeping up. If human activities related to ignitions or suppression change in the future (e.g., California's new ALERTCalifornia camera network instantaneously identifies fires across the vast majority of the state; https://alertcalifornia.org), then the WULFFSS model in its current formulation will become increasingly inaccurate. A possible option is to include a predictor variable such as gross domestic product (or perhaps state- and county-level median annual income), which is used in some earth-system modeling

schemes to serve as a proxy for technology available to aid suppression (Li et al., 2024a).

Finally, in developing WULFFSS we made the unconventional choice to bin separately the effects of predictors whose variance lies primarily in one of three domains: spatial, mean annual cycle, and lower-frequency temporal. Our reasoning was that spatial variations in the potential for fires to ignite and spread modulate the fire-promoting potency of temporal variations in

weather and climate. For example, climate conditions that dry out fuels are more likely to translate to heighted potential for wildfire in areas where fuels and potential ignition sources are abundant. However, the logic behind separating out the effects of climate into those driven by the mean annual cycle versus lower-frequency anomalies is debatable. On one hand, there is probably not a major difference between the mechanisms that cause wildfire activity to exhibit an annual cycle versus those that cause interannual variability, so allowing the model to represent these sources of variability as separate mechanisms is not

ideal. On the other hand, many climate variables share a similar annual cycle and climatological differences between opposing ends of the annual cycle are often much larger than the range of climatic variability that distinguishes years of high versus low fire potential. Thus, a statistical fire model trained on both intra- and inter-annual climate variability simultaneously risks over-representing variables that best correlate with the mean annual cycle in fire occurrences or sizes (e.g., solar intensity) but are not dominant drivers of interannual variability. That bias would dampen lower-frequency variability in simulated fire activity

and inhibit the diagnosis of past and future changes in western US forest-fire activity. High-quality data on live and dead fuel moistures could ameliorate the need to simulate the drivers of intra- and inter-annual variability separately by reducing our reliance on the multiple and covarying climate predictors that we currently use to represent the water balance (e.g., precipitation total, wet-day frequency, and *VPD* over multiple time scales).

**7 Conclusions**



We developed a monthly stochastic forest-fire model, WULFFSS, for the western US that operates on a 12-km resolution grid and simulates the probabilities and sizes of large fires (≥1 km² forest area burned). Predictor variables include vegetation characteristics, topography, human population, and climate. When trained with observed data WULFFSS reliably reproduces observed spatiotemporal variations in fire occurrence and area burned. Model performance remains high when tested in cross-validations against out-of-sample observations. The complex nature of wildfire and its nonlinear responses to many interacting

variables has motivated efforts to model wildfire with machine-learning techniques (Wang et al., 2021; Brown et al., 2023; Buch et al., 2023; Li et al., 2024b). These efforts are valuable, but should not wholly replace models that use conventional statistical methods that are generally more straight-forward to interpret and understandable by more people. Models developed using relatively simple methods provide value by establishing baselines against which machine-learning efforts can be compared. Further, it is increasingly evident that fire needs to be simulated within ecosystem and hydrological models in order

for plausible simulations of future changes to ecosystem composition, terrestrial carbon storage, snowpack, and streamflow (Bowman et al., 2009; Anderegg et al., 2022; Koshkin et al., 2022; Williams et al., 2022). Statistical modeling approaches therefore remain valuable in wildfire science, as ecosystem and land-surface modeling groups may be hesitant to adopt a machine-learning based fire model that is difficult to implement or explain. In the case of WULFFSS, we developed it to be coupled with our western US dynamical forest-ecosystem model, DYNAFFOREST (Hansen et al., 2022). With WULFFSS

and DYNAFFOREST, we can efficiently perform large ensembles of tens or hundreds of century-scale simulations of the coupled forest and wildfire processes across the western US. With this coupled approach we can quantitatively address questions about the relative contributions of human-caused climate change and fire-management practices to recent increases in forest-fire activity, how these importances have varied geographically, and how forest ecosystems and western US fire regimes may evolve under future climate change. Further, fire research is often heavily focused on fire frequency and size

because these metrics are easiest to observe. Coupling WULFFSS with a forest-ecosystem model allows us to investigate the drivers and rates of past and future trends of other important fire metrics such as severity and biomass loss. Finally, WULFFSS is a long-term, evolving project. Improvements will include simulation of fire spread, simulation of multiple tree cohorts to simulate ladder-fuel effects, simulation of grass and shrub communities to better represent fuel continuity, distinguishing between human versus natural fire ignitions, and explicit simulation of human effects on ignitions and fire sizes via

suppression.

**Appendices**

**Table A1.** Potential predictor variables dominated by spatial variability. "P model use" indicates whether the sign the effect of a given variable on fire probability had to be positive (+) or negative (-), or if a given variable was not considered as a potential predictor of fire probability (X). "Size model use" is same as "P model use" but for the fire-size model. "Round 2

only" indicates variables (X) only considered in the second round of model fitting. Variables with "in50kha" represent average conditions within a surrounding area of approximately 50,000 ha (a 23 km x 23 km box). Variables with "log10" are log-transformed. See Supplementary Table S1 for variable descriptions.





| Number | Name | P model use | Size model use | Round 2 only |
|---|---|---|---|---|
| V1_space | connectivity | + | + | |
| V2_space | connectivity_log10 | + | + | |
| V3_space | connectivity_in50kha | X | + | |
| V4_space | connectivity_in50kha_log10 | X | + | |
| V5_space | forestfrac | + | + | |
| V6_space | forestfrac_log10 | + | + | |
| V7_space | forestfrac_in50kha | X | + | |
| V8_space | forestfrac_in50kha_log10 | X | + | |
| V9_space | livebiomass_total | + | + | |
| V10_space | livebiomass_total_log10 | + | + | |
| V11_space | livebiomass_total_in50kha | X | + | |
| V12_space | livebiomass_total_in50kha_log10 | X | + | |
| V13_space | deadbiomass_total | + | + | |
| V14_space | deadbiomass_total_log10 | + | + | |
| V15_space | deadbiomass_total_in50kha | X | + | |
| V16_space | deadbiomass_total_in50kha_log10 | X | + | |
| V17_space | biomass_total | + | + | |
| V18_space | biomass_total_log10 | + | + | |
| V19_space | biomass_total_in50kha | X | + | |
| V20_space | biomass_total_in50kha_log10 | X | + | |
| V21_space | livebiomass_coarse | + | + | |
| V22_space | livebiomass_coarse_log10 | + | + | |
| V23_space | livebiomass_coarse_in50kha | X | + | |
| V24_space | livebiomass_coarse_in50kha_log10 | X | + | |
| V25_space | livebiomass_fine | + | + | |
| V26_space | livebiomass_fine_log10 | + | + | |
| V27_space | livebiomass_fine_in50kha | X | + | |
| V28_space | livebiomass_fine_in50kha_log10 | X | + | |
| V29_space | deadbiomass_coarse | + | + | |
| V30_space | deadbiomass_coarse_log10 | + | + | |

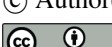



| | | | | |
|---|---|---|---|---|
| V31_space | deadbiomass_coarse_in50kha | X | + | |
| V32_space | deadbiomass_coarse_in50kha_log10 | X | + | |
| V33_space | deadbiomass_fine | + | + | |
| V34_space | deadbiomass_fine_log10 | + | + | |
| V35_space | deadbiomass_fine_in50kha | X | + | |
| V36_space | deadbiomass_fine_in50kha_log10 | X | + | |
| V37_space | cohort_dbh | + | + | |
| V38_space | cohort_dbh_log10 | + | + | |
| V39_space | cohort_dbh_in50kha | X | + | |
| V40_space | cohort_dbh_in50kha_log10 | X | + | |
| V41_space | cohort_height | + | + | |
| V42_space | cohort_height_log10 | + | + | |
| V43_space | cohort_height_in50kha | X | + | |
| V44_space | cohort_height_in50kha_log10 | X | + | |
| V45_space | spi_17to6monthsbefore_grass_shrub | | | |
| V46_space | spi_23to12monthsbefore_grass_shrub | | | |
| V47_space | spi_29to18monthsbefore_grass_shrub | | | |
| V48_space | spi_35to24monthsbefore_grass_shrub | | | |
| V49_space | unburnable | - | - | X |
| V50_space | unburnable_log10 | - | - | X |
| V51_space | unburnable_in50kha | X | - | X |
| V52_space | unburnable_in50kha_log10 | X | - | X |
| V53_space | agriculture | | - | X |
| V54_space | agriculture_log10 | | - | X |
| V55_space | agriculture_in50kha | X | - | X |
| V56_space | agriculture_in50kha_log10 | X | - | X |
| V57_space | developed | | | |
| V58_space | developed_log10 | | | |
| V59_space | developed_in50kha | X | | |
| V60_space | developed_in50kha_log10 | X | | |
| V61_space | slope | | + | |



| V62_space | slope_log10 | | + | |
|-----------|-------------|---|---|---|
| V63_space | elevstd | | | |
| V64_space | elevstd_log10 | | | |
| V65_space | aridityindex | - | - | |
| V66_space | aridityindex_log10 | - | - | |
| V67_space | hdwimaxann | + | + | |
| V68_space | hdwimaxann_log10 | + | + | |
| V69_space | fm1000 | - | - | |
| V70_space | fm1000_log10 | - | - | |
| V71_space | fm100 | - | - | |
| V72_space | fm100_log10 | - | - | |
| V73_space | seasindex | + | + | |
| V74_space | seasindex_log10 | + | + | |
| V75_space | lightning | + | X | |
| V76_space | lightning_log10 | + | X | |
| V77_space | housedensity | | | |
| V78_space | housedensity_log10 | | | |
| V79_space | housedensity_in50kha | X | | |
| V80_space | housedensity_in50kha_log10 | X | | |
| V81_space | dist5hpkm | | | |
| V82_space | dist5hpkm_log10 | | | |
| V83_space | dist50hpkm | | | |
| V84_space | dist50hpkm_log10 | | | |
| V85_space | fracsnow_1month | - | - | X |
| V86_space | fracsnow_log10_1month | - | - | X |
| V87_space | fracsnow_1monthafter | - | - | X |
| V88_space | fracsnow_log10_1monthafter | - | - | X |
| V89_space | swemean_1month | - | - | X |
| V90_space | swemean_log10_1month | - | - | X |
| V91_space | swemean_1monthafter | - | - | X |
| V92_space | swemean_log10_1monthafter | - | - | X |



| | | | | |
|---|---|---|---|---|
| V93_space | roaddist_major | | | X |
| V94_space | roaddist_major_log10 | | | X |
| V95_space | roaddist_minor | | | X |
| V96_space | roaddist_minor_log10 | | | X |
| V97_space | roaddist_all | | | X |
| V98_space | roaddist_all_log10 | | | X |

**Table A2.** As in Table A1 but potential predictor variables representing the mean annual cycle. Climate predictors with "mean" indicate the mean annual cycle of monthly values during the model calibration period (1985–2024). Variables with "mean_scaled" are mean annual cycles linearly scaled between zero for the annual minimum and 1 for the annual maximum. Durations at the end of variable names (e.g., "3 month") indicate that monthly values were averaged with a moving window of the indicated duration prior to calculation of the annual cycle, with the moving window ending in the month for which the average was assigned (e.g., the 3-month average assigned to March is calculated across January–March). Variables with "1monthafter" represent mean climate of the next month (e.g., the mean annual cycle value assigned to January represents that of February). See Supplementary Table S2 for variable descriptions.

| Number | Name | P model use | Size model use | Round 2 only |
|---|---|---|---|---|
| V1_seas | aridityindex_mean_scaled_1month | - | - | |
| V2_seas | aridityindex_mean_scaled_2month | - | - | |
| V3_seas | aridityindex_mean_scaled_3month | - | - | |
| V4_seas | aridityindex_mean_scaled_1monthafter | X | - | |
| V5_seas | prec_mean_scaled_1month | - | - | |
| V6_seas | prec_mean_scaled_2month | - | - | |
| V7_seas | prec_mean_scaled_3month | - | - | |
| V8_seas | prec_mean_scaled_1monthafter | X | - | |
| V9_seas | wetdays_mean_scaled_1month | - | - | |
| V10_seas | wetdays_mean_scaled_2month | - | - | |
| V11_seas | wetdays_mean_scaled_3month | - | - | |
| V12_seas | wetdays_mean_scaled_1monthafter | X | - | |
| V13_seas | vpd_mean_scaled_1month | + | + | |
| V14_seas | vpd_mean_scaled_2month | + | + | |
| V15_seas | vpd_mean_scaled_3month | + | + | |
| V16_seas | vpd_mean_scaled_1monthafter | X | + | |
| V17_seas | solar_mean_scaled_1month | + | + | |
| V18_seas | solar_mean_scaled_2month | + | + | |





| V19_seas | solar_mean_scaled_3month | + | + | |
|----------|--------------------------|---|---|---|
| V20_seas | solar_mean_scaled_1monthafter | X | + | |
| V21_seas | cape_mean_scaled_1month | + | + | |
| V22_seas | lightning_mean_scaled_1month | + | X | |
| V23_seas | hdwi_mean_scaled_1month | + | + | |
| V24_seas | hdwi_mean_scaled_1monthafter | X | + | |
| V25_seas | hdwi_max1day_scaled_1month | + | + | |
| V26_seas | hdwi_max1day_scaled_1monthafter | X | + | |
| V27_seas | hdwi_max3day_scaled_1month | + | + | |
| V28_seas | hdwi_max3day_scaled_1monthafter | X | + | |
| V29_seas | aridityindex_log10_mean_1month | - | - | |
| V30_seas | aridityindex_log10_mean_2month | - | - | |
| V31_seas | aridityindex_log10_mean_3month | - | - | |
| V32_seas | aridityindex_log10_mean_1monthafter | X | - | |
| V33_seas | fm1000_mean_scaled_1month | - | - | |
| V34_seas | fm1000_mean_scaled_1monthafter | X | - | |
| V35_seas | fm100_mean_scaled_1month | - | - | |
| V36_seas | fm100_mean_scaled_1monthafter | X | - | |
| V37_seas | prec_log10_mean_1month | - | - | |
| V38_seas | prec_log10_mean_2month | - | - | |
| V39_seas | prec_log10_mean_3month | - | - | |
| V40_seas | prec_log10_mean_1monthafter | X | - | |
| V41_seas | wetdays_mean_1month | - | - | |
| V42_seas | wetdays_mean_2month | - | - | |
| V43_seas | wetdays_mean_3month | - | - | |
| V44_seas | wetdays_mean_1monthafter | X | - | |
| V45_seas | vpd_mean_1month | + | + | |
| V46_seas | vpd_mean_2month | + | + | |
| V47_seas | vpd_mean_3month | + | + | |
| V48_seas | vpd_mean_1monthafter | X | + | |
| V49_seas | solar_mean_1month | + | + | |





| V50_seas | solar_mean_2month | + | + | |
|---|---|---|---|---|
| V51_seas | solar_mean_3month | + | + | |
| V52_seas | solar_mean_1monthafter | X | + | |
| V53_seas | cape_mean_1month | + | + | |
| V54_seas | lightning_mean_1month | + | X | |
| V55_seas | hdwi_mean_1month | + | + | |
| V56_seas | hdwi_mean_1monthafter | X | + | |
| V57_seas | hdwi_max1day_1month | + | + | |
| V58_seas | hdwi_max1day_1monthafter | X | + | |
| V59_seas | hdwi_max3day_1month | + | + | |
| V60_seas | hdwi_max3day_1monthafter | X | + | |
| V61_seas | fm1000_mean_1month | - | - | |
| V62_seas | fm1000_mean_1monthafter | X | - | |
| V63_seas | fm100_mean_1month | - | - | |
| V64_seas | fm100_mean_1monthafter | X | - | |
| V65_seas | vpdmax_mean_scaled_1month | + | + | X |
| V66_seas | vpdmax_mean_scaled_2month | + | + | X |
| V67_seas | vpdmax_mean_scaled_3month | + | + | X |
| V68_seas | vpdmax_mean_scaled_1monthafter | X | + | X |
| V69_seas | vpdmin_mean_scaled_1month | + | + | X |
| V70_seas | vpdmin_mean_scaled_2month | + | + | X |
| V71_seas | vpdmin_mean_scaled_3month | + | + | X |
| V72_seas | vpdmin_mean_scaled_1monthafter | X | + | X |
| V73_seas | vpdmax_mean_1month | + | + | X |
| V74_seas | vpdmax_mean_2month | + | + | X |
| V75_seas | vpdmax_mean_3month | + | + | X |
| V76_seas | vpdmax_mean_1monthafter | X | + | X |
| V77_seas | vpdmin_mean_1month | + | + | X |
| V78_seas | vpdmin_mean_2month | + | + | X |
| V79_seas | vpdmin_mean_3month | + | + | X |
| V80_seas | vpdmin_mean_1monthafter | X | + | X |



**Table A3.** As in Table A1 but potential predictor variables represent temporal variability at timescales beyond the annual cycle. Durations at the end of variable names (e.g., "3 month") indicate that monthly values were averaged with a moving window of the indicated duration prior to calculation of anomalies, with the moving window ending in the month for which the average was assigned (e.g., the 3-month average assigned to March is calculated across January–March). Variables with "1monthafter" represent mean climate of the next month (e.g., the mean annual cycle value assigned to January represents that of February). Variables with "anom" are standardized such that, for each of the 12 months, the mean is zero and standard deviation is 1 during the model calibration period of 1985–2024. See Supplementary Table S3 for variable descriptions.

| Number | Name | P model use | Size model use | Round 2 only |
|---|---|---|---|---|
| V1_temporal | spi_1month | - | - | |
| V2_temporal | spi_2month | - | - | |
| V3_temporal | spi_3month | - | - | |
| V4_temporal | spi_4month | | | |
| V5_temporal | spi_5month | | | |
| V6_temporal | spi_6month | | | |
| V7_temporal | spi_9month | | | |
| V8_temporal | spi_12month | | | |
| V9_temporal | spi_1monthafter | X | - | |
| V10_temporal | spi_17to6monthsbefore | | | |
| V11_temporal | spi_23to12monthsbefore | | | |
| V12_temporal | spi_29to18monthsbefore | | | |
| V13_temporal | spi_35to24monthsbefore | | | |
| V14_temporal | wetdays_anom_1month | | - | |
| V15_temporal | wetdays_anom_2month | | - | |
| V16_temporal | wetdays_anom_1monthafter | X | - | |
| V17_temporal | vpd_anom_1month | + | + | |
| V18_temporal | vpd_anom_2month | + | + | |
| V19_temporal | vpd_anom_3month | + | + | |
| V20_temporal | vpd_anom_4month | + | + | |
| V21_temporal | vpd_anom_5month | + | + | |
| V22_temporal | vpd_anom_6month | + | + | |
| V23_temporal | vpd_anom_9month | + | + | |
| V24_temporal | vpd_anom_12month | + | + | |



| | | | | |
|---|---|---|---|---|
| V25_temporal | vpd_anom_1monthafter | X | + | |
| V26_temporal | cape_anom_1month | + | + | |
| V27_temporal | cape_anom_max1day_1month | + | + | |
| V28_temporal | cape_anom_max3day_1month | + | + | |
| V29_temporal | hdwi_anom_1month | + | + | |
| V30_temporal | hdwi_anom_1monthafter | X | + | |
| V31_temporal | hdwi_anom_max1day_1month | + | + | |
| V32_temporal | hdwi_anom_max1day_1monthafter | X | + | |
| V33_temporal | hdwi_anom_max3day_1month | + | + | |
| V34_temporal | hdwi_anom_max3day_1monthafter | X | + | |
| V35_temporal | fm1000_anom_1month | - | - | |
| V36_temporal | fm1000_anom_1monthafter | X | - | |
| V37_temporal | fm1000_anom_min3day_1month | - | - | |
| V38_temporal | fm1000_anom_min3day_1monthafter | X | - | |
| V39_temporal | fm100_anom_1month | - | - | |
| V40_temporal | fm100_anom_1monthafter | X | - | |
| V41_temporal | fm100_anom_min3day_1month | - | - | |
| V42_temporal | fm100_anom_min3day_1monthafter | X | - | |
| V43_temporal | vpd_anom_max3day_1month | + | + | |
| V44_temporal | vpd_anom_max3day_1monthafter | X | + | |
| V45_temporal | fracsnow_anom_1month | - | - | |
| V46_temporal | fracsnow_anom_1monthafter | X | - | |
| V47_temporal | swemax_last12months_anom | - | - | |
| V48_temporal | swemax_last12months | - | - | X |
| V49_temporal | fracsnow_1month | - | - | X |
| V50_temporal | fracsnow_log10_1month | - | - | X |
| V51_temporal | fracsnow_1monthafter | X | - | X |
| V52_temporal | fracsnow_log10_1monthafter | X | - | X |
| V53_temporal | swemean_1month | - | - | X |
| V54_temporal | swemean_log10_1month | - | - | X |
| V55_temporal | swemean_1monthafter | X | - | X |





| V56_temporal | swemean_log10_1monthafter | X | - | X |
|---|---|---|---|---|
| V57_temporal | vpdmax_anom_1month | + | + | X |
| V58_temporal | vpdmax_anom_1monthafter | X | + | X |
| V59_temporal | vpdmin_anom_1month | + | + | X |
| V60_temporal | vpdmin_anom_1monthafter | X | + | X |
| V61_temporal | vpdmax_anom_max1day_1month | + | + | X |
| V62_temporal | vpdmax_anom_max1day_1monthafter | X | + | X |
| V63_temporal | vpdmax_anom_max3day_1month | + | + | X |
| V64_temporal | vpdmax_anom_max3day_1monthafter | X | + | X |
| V65_temporal | vpdmin_anom_max1day_1month | + | + | X |
| V66_temporal | vpdmin_anom_max1day_1monthafter | X | + | X |
| V67_temporal | vpdmin_anom_max3day_1month | + | + | X |
| V68_temporal | vpdmin_anom_max3day_1monthafter | X | + | X |

**Code and data availability**

The datasets and the code used to produce the WULFFSS are available for peer review at
http://datadryad.org/share/Ox4oxdwdrhkmjUTpke7QgkfF--h-RLRbmMzGBhSmOr4
https://doi.org/10.5061/dryad.63xsj3vdb (Williams, 2025)

**Author contributions**

APW, WDH, and JB conceptualized the research. APW and WDH designed the methodology and performed the analyses. APW, WDH, CSJ, JTA, VCR, and BW curated the data. APW and JH performed validation. APW wrote the original manuscript draft. All authors edited the manuscript. WDH and AWP acquired the funding that supported this work.

**Competing interests**

The authors declare that they have no conflict of interest.

**Acknowledgments**



This work was improved by insights from members of the Western fire & Forest Resilience Collaborative (https://www.westernfireforest.org/) as well as use of the Derecho supercomputer, provided by the National Science Foundation and the National Center for Atmospheric Research (https://www.cisl.ucar.edu/capabilities/derecho). We thank Mark Handcock for helping us optimize our approach to model parameterization.

## Financial support

This research was supported by the Zegar Family Foundation, the John D. and Catherine T. MacArthur Foundation, the Gordon and Betty Moore Foundation (11974 and 13283), the UCLA Sustainable LA Grand Challenge, and the USGS Southwestern Climate Adaptation Science Center (G24AC00611 and G24AC00080).

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
