# Peer review of "The Western United States Large Forest-Fire Stochastic Simulator (WULFFSS) 1.0: A monthly gridded forest-fire model using interpretable statistics"

_EGUsphere, 2025_

## Author Response (AR1)

To the editor and reviewers: In addition to the revisions we made in light of reviewer comments, we have updated the fire model using an updated version of the WUMI2024a database of observed wildfires as well as updated climate data. The paper describing the WUMI2024a database has now been accepted at *Earth System Science Data* and the revisions to the database were made in response to reviewer comments on that paper. The revisions to the climate datasets were made because the version of the dynamically downscaled ERA5 climate dataset that we used previously ended in August 2023 but that dataset has since been updated through April 2025. So, in all cases where we previously extended the dynamically-downscaled ERA5 data with bias-corrected values from other datasets, we now use the dynamically-downscaled ERA5 for the full study period for consistency. These improvements to the observed fire and climate datasets did not cause major changes to the model but nonetheless represent an improvement in the approach.

**Reviewer #1**
Summary:

This paper introduces WULFFSS, a newly developed statistical model designed to simulate the monthly probability and size of large forest fires (size ≥100 ha) across the western US, on a 12-km spatial resolution. The modeling framework consists of 3 statistical models that are called stepwise, forced by climate, vegetation, topographic, and anthropogenic variables. Two of the three models each consist of three components representing different groups of predictor variables, as well as interactions in between them. The modeling framework features its stochastic nature and interpretability, and computational efficiency. The framework is designed to be couple with forest ecosystem model (such as DYNAFFOREST) to allow bidirectional feedbacks between the vegetation and fire events. The application of the framework for 1985-2024 is able to capture the majority interannual variability of forest fires in the studied western US with some limitation.

Comments:

The authors did a good job in building the statistical modeling framework to estimate fire probability and sizes in western US from multiple influencing groups of variables. Allowing the model to couple with an forest ecosystem model for feedbacks indeed promotes its future potential. However, there are questions about methodological choices and performance analysis that further explanation needs to be made to clarify the robustness of this paper.

We thank the reviewer for their thorough and constructive review. We are grateful in particular for the motivation to include GDP as a predictor variable and to dig further into the model's underperformance in the CA/NV region. We have revised the paper in light of the reviewer's comments/suggestions and we provide point-by-point responses in blue font below.

1. The modeling framework consists of three stepwisely connected statistical models, how sensitive is the final simulation to the prior step variables?

The model's behavior would definitely be sensitive to changes in the order in which the spatial, seasonal, and temporal components are constructed. This is by design. We describe the logic behind our approach in response to the reviewer's next comment, as well as a revision we made to the text.

2. For model P and A, each consists of the three component groups of predictor variables. The variables are categorized into spatial, annual cycle and temporal domains to be examined. This is quite different from the more conventional method. Why took this approach, instead of treating all predictors in the same single pool? Is there quantitative consideration that the three-domain method better than the standard multi-variable regression?

The order in which each model is developed, first spatial, then seasonal, then temporal is motivated by our understanding that factors related to fuel availability and ignitions, which are highly variable spatially, strongly modulate the influence of climate. In particular, if fuel variables are not accounted for first, then a regression of fire activity against variables related to moisture/aridity would likely indicate a hump-shaped response, where fire appears to be promoted by additional moisture among places where moisture average balance is relatively dry or moderate (e.g., Bradstock 2010; Krawchuk and Moritz, 2011). However, this positive response of fire to moisture is due to the positive effect of moisture on fuel abundance, so accounting for the positive effect of fuels at the outset allows for more accurate subsequent detection of the negative effect of moisture on fire via reductions in flammability.

Further, in statistical analyses of the spatiotemporal drivers of fire across large and diverse regions throughout the year, variability in fire activity is far greater across space and seasonally than interannually. Thus, a statistical modeling approach aimed at simply minimizing errors would prioritize predictors that correlate best with fire variability in the spatial and seasonal domains. However, a main motivation of our work is to develop a model that can be used to explore temporal changes in fire activity at timescales of interannual and longer. This explains our choice to account for variations in fire activity associated with the mean annual climate cycle before accounting for variations at longer temporal scales.

Notably, we did acknowledge the uniqueness of our 3-component modeling approach and describe the logic behind it in the final paragraph of section 6 (Discussion, strengths, and limitations) or our original submission. That paragraph remains in our revised paper and we also now better describe the logic behind our unique modeling approach when it is first introduced in section 4 (Model description). The new text reads:

L394–401 "*The S component is constructed first to capture the how variations in fire activity are driven by factors that are far more variable in space than in time, as these factors (e.g., forest biomass, lightning frequency, variables related to human population and fire suppression) are likely to modulate the sensitivity of fire activity to temporal variables. The C component is then constructed to account for variations in fire activity that are due to the mean annual climate cycle. Finally, the T component is constructed to account for effects of interannual climate variability, which are likely to be strongly modulated by the effects of the S and C variables already accounted for.*"

Bradstock, R. A. "A Biogeographic Model of Fire Regimes in Australia: Current and Future Implications." *Global Ecology and Biogeography* 19, no. 2 (2010): 145–58. https://doi.org/10.1111/j.1466-8238.2009.00512.x.

Krawchuk, M. A., and M. A. Moritz. "Constraints on Global Fire Activity Vary across a Resource Gradient." *Ecology* 92, no. 1 (2011): 121–32. https://doi.org/10.1890/09-1843.1.

3. When presenting the landcover data in the model, the author uses 41 grid cells from 8 directions surrounding the target grid cell to calculate the forest connectivity. Is there specific reason how the chosen area (or, the number of the grid cells) is set?
This question pertains to our calculation of the forest connectivity variable, which we calculate at a 1-km resolution based on connectedness of forest in the nearest 6 grid cells radiating from the central grid cell in the 4 cardinal directions and the 4 grid cells radiating in 4 the diagonal directions. The logic behind these numbers is that we wanted the area represented to be aligned with that of a fairly large fire, approximately 10,000 ha. We also wanted the connectivity metric to be a fast calculation so that it can be easily recalculated at the 1-km scale on an annual basis in coupled simulations where a forest-ecosystem model is interacting with the fire model and the landcover metrics are constantly being recalculated. Thus, we chose to represent connectivity in a relatively simple way, considering just the 8 directions radiating from the central grid cell rather than across all grid cells in a circle. In this case, a circular area represented on a 1-km grid and centered symmetrically around a central 1-km grid cell would have a diameter of 11-13 grid cells in the cardinal directions and 9 cells in the diagonals. We have added the following sentence to section 3.4 (Landcover) to clarify the logic:

L334–339: "*This approach allows for efficient recalculations of connectivity when simulations are run in coupled mode with DYNAFFOREST and the size of the area represented is roughly aligned with that of a large wildfire 10,000 ha in size.*"

4. When introducing the human factor in the model, the author uses population density and distance to road, as predictors. Could the author please explain why some other widely used predictors, such as GDP or fire agency availability, are not included in the domain, relating to suppression capability?
This question motivated us to add GDP into the list of potential spatial predictors considered. In the Discussion section (2nd to last paragraph) we had previously indicated this as a variable that we should consider in future. In the current model-building process we consider the 0.5-degree maps of annual GDP and GDP per capita produced by Kummu et al. (2025), as well as log-transformations of these variables. However, the GDP variables were not selected as predictors of fire probability or size in our step-wise model building process, as distance from populations and roads provided more model skill. The reason why GPD did not contribute important unique information about fire probability and size may be that federal suppression resources are used for many forest fires, which may work to dampen the effects of regional variability in wealth. We believe finer-scale features related to local availability of suppression resources, as the reviewer suggests, may be more important determinants of suppression capacity and effectiveness but we have not yet identified suitable databases that would allow us to represent how local resources, for example, fire-station density or availability of nearby aircraft for aerial attack, have changed over our study period.

In addition to the additional Methods text describing the GDP data we have revised the portion of the Discussion section that previously indicated how future modeling efforts should consider use of GDP as a predictor:

L1036–1041: "*Variables more directly related to suppression capacity than population and road density may be helpful in future modelling efforts. Notably, our use of annual maps of gross domestic product, a variable used in some earth-system modeling schemes to serve as a proxy for suppression capacity (Li et al., 2024a), did not contribute to model skill. Federal suppression resources may make up for much of the regional variability in wealth. Finer-scale features such as distance to the nearest fire station or aircraft availability for aerial firefighting may prove valuable in future efforts.*"

Kummu, M., M. Kosonen, and S. Masoumzadeh Sayyar. "Downscaled Gridded Global Dataset for Gross Domestic Product (GDP) per Capita PPP over 1990–2022, Scientific Data, 12, 178." *Scientific Data* 12 (2025): 178. https://doi.org/10.1038/s41597-025-04487-x.

Li, F., X. Song, S. P. Harrison, et al. "Evaluation of Global Fire Simulations in CMIP6 Earth System Models." *Geoscientific Model Development Discussions* 17 (2024): 1–37. https://doi.org/10.5194/gmd-2024-85.

5. The model yields quite good simulations for the other regions in the western US, yet it evidently underperforms in the CA/NV region on both annual frequency and forest area burned (Fig. 16d and Fig. 17d). Apart from the potential reasons listed, did author analyse as to where the discrepancy comes from, was it mainly caused by ignition probability, or together with fire size? What is the dominant driver for this bias?

This reviewer comment motivated us to take a deeper dive by evaluating model performance at the sub-regional level. New supplementary Figures S1 and S2 that show observed versus modeled time series of fire frequency and area burned for each of the 11 western US states. In Figure R1 below we show just the California and Nevada panels relevant to this question.

First, fire frequency rather than fire size is the main driver of the model underperformance in the CA/NV region, as correlations between observed and modeled fire frequencies in CA/NV are substantially worse than in the other regions, while correlations for area burned are in better alignment with the other regions.

Of the two states, California has approximately four times more forest fires and approximately 17 times more forest area burned than Nevada, so the CA/NV time series shown in Figs 16 and 17 (now 17 and 18) are dominated by California, and California is mostly responsible for the poor correlation between observed versus modeled fire frequency. The low correlation for fire frequency in California is driven in part by very high fire counts in 1987 and 2008 caused by anomalous outbreaks of dry lightning that are not adequately represented in our modeling approach. We mentioned this in the previous submission, but stress this limitation more explicitly in the revision (more on this below), and now attribute these problematic years specifically to California in light of our state-specific analysis. We also noted in the previous submission that the model overestimated CA/NV fire frequencies in 2021, 2022, and 2024 and we maintain our explanation that these over-estimates may be due to a combination of factors including increased suppression capacity and public/corporate awareness of fire hazards as well as fire-and drought-related reductions in fuel continuity that our modeling may not capture.

Nonetheless, the new Figure S2 and the right side of Fig. R1 below show that simulations of total area burned in California are quite skillful, more so even than in a number of other states.

Even though California dominates the CA/NV regional record, our state-by-state analysis highlights that model limitations in Nevada also reduce skill in CA/NV. Our model systematically overestimates fire frequencies in Nevada by approximately 70%, and then overestimates area burned by nearly a factor of four. We believe there are two main reasons for this. First, we believe our model underrepresents the role that fuel limitation plays in Nevada's relatively sparse and narrow forested corridors. This is likely due to positive biases in the fuel loads and fuel connectivity simulated by our DYNAFFOREST model, and a related under-representation of the importance of climate-driven interannual variations in non-forest fuels. In addition, WULFFSS only detects a positive association between distance from humans and fire frequency, but this fails to adequately represent areas like the remotest forested areas of Nevada where there is little-to-no human footprint and thus essentially zero human-caused fires. Of these two limitations, we believe the fuel-limitation issue is more consequential to our simulations in Nevada, as the positive bias in simulated fire activity becomes even more serious when it comes to fire size. Our model simulates nearly four times more area burned in Nevada than is observed. When it comes to fire sizes, the effect of remoteness from human access should only be positive, and yet observed fires are systematically smaller than those that we simulate in Nevada.

In light of the above, we have made a number of a revisions to the paper:

In the part of section 5 (Model performance) that describes the skill of our fire-frequency estimates we have greatly expanded our description and explanation of the relatively low skill in CA/NV:

L851–861: "*Reasons for model underperformance in CA/NV are numerous. In California (Fig. S1c), the large observed fire frequencies in 1987 and 2008 were due to anomalous dry lightning events (Kalashnikov et al., 2022), which are not adequately represented in WULFFSS. The more recent overestimates in California fire frequency may be due to increased resources for fire detection and suppression in California, increased public and corporate awareness of fire hazards, and reductions in fuel continuity due to drought and related bark-beetle outbreaks that our modeling does not capture. Nevada also contributes to the relatively low model skill in CA/NV (Fig. S1g); WULFFSS overestimates mean fire frequency by approximately 70% in Nevada, a far larger mean bias than for any other state. The overestimates of fire activity in Nevada's sparse and isolated Great Basin forests suggest that our approach underestimates the ability of low biomass and vegetation connectivity to limit fire activity and/or that our DYNAFFOREST-based estimates of biomass and connectivity are too high there. In addition, while our model indicates that fire frequency is positively related to remoteness from human population (Fig. 5), ignitions may be a limiting factor in forested areas of Nevada with especially light human footprints.*"

In the next part of section 5 describing the skill of our area-burned estimates we have added:

L896–901: "*In our state-specific analysis we find that overestimates of area burned in CA/NV are apparent in both California and Nevada, but the bias is more severe and systematic in*

*Nevada, where WULFFSS models nearly four times more area burned than is observed (Fig. S2). This is the largest such bias among the 11 states, followed by Utah where estimates of area burned exceed observations by a factor of two. Consistent overestimates of area burned in these states, home to the relatively dry and spatially discontinuous forests of the Great Basin, further implicates fuel limitation in sparsely forested areas as a cause of error for WULFFSS."*

Finally, we expanded on the fuel-limitation issue highlighted by the Nevada biases in the Discussion section:

L989–997: *"This limitation appears most clearly in our simulation of fire in the isolated forests atop the narrow and arid mountain ranges of the Great Basin. In Nevada, for example, WULFFSS overestimates fire frequency by 70% and area burned by a factor of four. In addition to limitations caused by our current lack of representation of non-forest fuel dynamics, overestimates of Great Basin fire activity are also probably promoted by positive biases in our DYNAFFOREST-simulated maps of forest biomass and connectivity in the Great Basin region. This further motivates the need for spatially continuous maps of observed (or inferred from remotely sensed imagery) vegetation biomass across the western US that cover the time period of 1980s to near present at timesteps of annual or finer, which could be used as forcings in WULFFSS simulations of the observational period and to improve vegetation ecosystem models such as DYNAFFOREST."*

[Figure]

*Fig. R1. Alternative versions of Figures 17 and 18 in the main paper, but here focusing on the California/Nevada (CA/NV) region as well as California and Nevada separately. See the new Supplementary Figures S1 and S2 for similar plots for each of the 11 states in the western US domain.*

6. The model significantly underestimates the extreme fire events in year 2020 and 2024 (in terms of fire size), does this imply that the model systematically lack the ability to capture fire events induced by extreme weather conditions accurately? Can author suggest ways to improve this?

The underestimates of area burned in 2020 and 2024 are driven by the PNW and CA/NV

regions, though the 2020 estimate is improved some degree in CA/NV in the new submission and the underestimate in 2024 was and still is entirely driven by PNW. In our original submission we noted the 2020 and 2024 underestimates and discussed their likely causes. Below we describe how these underestimates were described in the original submission and we then describe a new paragraph that we have added to the Discussion section to further highlight what can be learned from these events.

2020: In our previous submission, section 5 (Model performance) described the combination of unique circumstances that were likely contributors to the underestimated area burned in 2020. In final paragraph of that section, we wrote: "*In 2020, however, WULFFSS grossly underestimates CA/NV area burned, and to a lesser extent in PNW (Fig. 17b,d). Potential contributing factors include rare lightning storms from tropical storm Fausto in August 2020, two extreme heat waves in the days to weeks immediately following the lightning storms, and overstretched suppression resources due to a high concentration of large forest fires in California and Oregon and the COVID-19 pandemic.*" In the Discussion section we then followed up with "*A likely explanation is that when a rare summertime lightning event coincided with hot and dry conditions to produce widespread wildfire activity, coupled with the COVID-19 pandemic, suppression efforts had difficulty keeping up.*"

Notably, the underestimates in CA/NV area burned in 2020 are somewhat improved in the revised version of the model because of the inclusion of convective available potential energy (*CAPE*) in the new fire-size model, which was observed to encourage extreme plume development and fire weather that promoted rapid fire growth in California in 2020 (e.g., Lee et al. 2023). The reason the new model incorporates *CAPE* is likely because of improved accuracy of our *CAPE* dataset. Previously the dynamically-downscaled version of daily ERA5 *CAPE* was only available through August 2023, so we had to use daily *CAPE* from the North American Regional Reanalysis to extend through 2024. Now, the downscaled ERA5 data are available through April 2025, allowing for our full model-calibration period to be covered by that high-quality dataset.

2024: As for the underestimates in area burned in 2024 in the PNW region, our original manuscript indicated in section 5 (Model performance) that this was largely driven by a large underestimate in the number of fires in that year. The 2024 forest-fire frequency in PNW was nearly highest on record and we indicated in the original submission that our underestimate was "*due to a failure to capture the large number of fires in Oregon and southwest Idaho that ignited from outbreaks of dry lightning in mid and late July.*" We went on to describe that our model does not use lightning as a predictor due to lack of an observational dataset that is free of major temporal inconsistencies and that, "*while CAPE is considered as a potential predictor of fire frequency due to its association with lightning, high CAPE is also associated with precipitation, limiting its value as a proxy for dry lightning.*"

A lesson from the above years, as well as some of the other shortcomings in the CA/NV region discussed in response to the reviewer's previous comment, is that the lack of explicit representation of lighting events is an important limitation. As we note in section 5, "*While WULFFSS does consider long-term mean patterns of lighting activity, it does not model fire as a function of temporal variability in lightning because the only long-term lighting dataset we are*

*aware of (from the NLDN, 1987–present) has temporal instabilities due to instrumental changes and it does not cover the full model-calibration period. While CAPE is considered to be a T variable due to its coincidence with lightning and atmospheric instability, high CAPE is also associated with precipitation, limiting its value as a proxy for dry lightning."*

In the new submission, we have added a new paragraph to the Discussion section that directly describes the limitations caused by lack of explicit representation of temporal variations in lightning frequency, and dry-lightning in particular:

L1043–1054: *"WULFFSS does not capture the important contributions of dry-lightning events, particularly near the west coast where lightning is relatively rare and thus a single anomalous event can cause a large increase in annual fire frequency and area burned. For example, the very high fire counts in CA/NV in 1987 and 2008 and in PNW in 2024 were due in part to anomalous outbreaks of dry lighting. Temporal variations in lightning frequency are not currently used as predictors in WULFFSS because we are not aware of an observational lightning dataset that spans our full model-calibration period and is not free of temporal inconsistencies due to changes in observational methods. Ideally, lightning would be a variable that can be modelled based on meteorological data, allowing lightning to force model simulations representing time periods or idealized scenarios beyond the 1985–2024 period of focus here. While lightning frequency has been shown previously to be well correlated to CAPE multiplied by precipitation total (Romps et al., 2018), the likelihood of ignition from lightning is substantially reduced if it coincides with precipitation. We thus consider CAPE on its own as a potential proxy for dry lightning potential, but ultimately CAPE was not selected by our fire-probability model. Future efforts to identify meteorological proxies for dry-lightning potential would likely enhance our model's simulations of fire-frequency extremes."*

In addition, we also now highlight at the end of the final paragraph of section 5 that some of the apparent underestimate in 2024 area burned is caused by an observational bias associated with a lack of MTBS data for that year. We now explain:

L907–913: *"In 2024, the large underestimate of fire frequency in PNW noted above (Fig. 17b), in Oregon and Idaho specifically (Fig. S1), translated to underestimates in total area burned (Figs. 18b, S2). However, it is likely that our observational record of area burned is biased high in 2024, as MTBS maps are not yet available for most large wildfires in that year, so the currently available maps of many of that year's largest fires do not represent within-fire spatial heterogeneity in area burned. On average, MTBS maps indicate that approximately 20% of area within forest-fire perimeters is unburned, so it is likely that our underestimate of area burned in 2024 will be lessened somewhat once MTBS data become available."*

Lee, J. M., J. D. Mirocha, N. P. Lareau, et al. "Sensitivity of Pyrocumulus Convection to Tree Mortality during the 2020 Creek Fire in California." *Geophysical Research Letters* 50, no. 16 (2023): e2023GL104193. https://doi.org/10.1029/2023GL104193.

Romps, D. M., A. B. Charn, R. H. Holzworth, W. E. Lawrence, J. Molinari, and D. Vollaro. "CAPE Times P Explains Lightning over Land but Not the Land-ocean Contrast." *Geophysical Research Letters* 45, no. 22 (2018): 12–623. https://doi.org/10.1029/2018GL080267.

7. By introducing self-regulation effect that reduces fuels from previous fires, will this "saturation" effect limit the model performance in future projection, considering that the climate change and ecosystem feedbacks may alter its self-regulation point?

If we understand the reviewer's question correctly, the desire to make plausible simulations of future fire under scenarios of changing fuel availability represents one of the main motivations behind the creation of this model specifically as a fire module that can be incorporated into an ecosystem model. That is, we have good reason to believe that, if forest fires continue to grow larger and more severe, they should also become increasingly self-regulating and thus relationships between climate and forest-fire metrics such as frequency or area burned should become weaker. Our modeling framework is specifically designed to allow for simulation of this process, as when coupled to an ecosystem model such as DYNAFFOREST, fuel characteristics are simulated to change in response to fire, and these changes then alter the predictor variables used in the fire model such that subsequent fire-climate relationships are modulated. This motivation was and still is explained in final three paragraphs of the Introduction and the 4th paragraph of the Discussion section.

**Reviewer #2**

General comments

This manuscript presents the development and evaluation of WULFFSS, a novel stochastic monthly gridded model for simulating large forest fires in the western United States. The model operates at a 12-km resolution and leverages interpretable statistical methods to estimate fire probability and size. Its key strengths include the integration of spatial, annual cycle and temporal anomaly components that can interact, high computational efficiency enabling large ensembles, and its design for coupling with the DYNAFFOREST ecosystem model to simulate vegetation-fire feedbacks. The model demonstrates considerable skill in capturing frequency and extent of western US forest fires, as validated through rigorous temporal and spatial cross-validation. The manuscript is thorough, well-structured, and describes the model, data, and validation procedures in significant detail. The work represents a valuable contribution to the field of fire modeling. However, the text is somewhat lengthy, and there is room for improvement. Therefore, I recommend that this manuscript be accepted with minor revisions.

We thank the reviewer for the thorough and constructive review. We are especially glad that this review motivated us to add a diagram of the general model structure and also to look into the effects of including 3-way interactions between the spatial, mean annual climate cycle, and temporal climate anomaly components of the model. Although including this term did not improve model performance and we thus chose to not keep it, we suspect this would have been a common question among readers and we now explicitly note that the 3-way interaction was considered but not included because it did not contribute additional skill. We also appreciate the reviewer's concern shared above regarding the length of the paper. In light of that comment we carefully went over the writing with conciseness in mind. The clearest opportunities for streamlining were in the Introduction and the revised Introduction is approximately 200 words shorter than the original. We also appreciate the reviewer's suggestion to include a diagram illustrating the general model framework and we now include such a figure. Notably, most reviewer requests were for additional details, so ultimately the revised paper is longer than the original despite our work to shorten the original text. We have revised the paper in light of the reviewer's comments and suggestions and we provide point-by-point responses in blue font below.

Specific comments

1. The model operates at a 12-km resolution, and it is possible that within a grid cell, the actual burned area constitutes only a small fraction. A key concern is the issue of intra-grid-cell heterogeneity and the representativeness of the predictor variables. It would be valuable if the authors could discuss the potential implications of this scaling issue.

We agree that the probability and size of a fire occurring in a given 12-km grid cell should be sensitive to heterogeneity at the sub-12-km scale. We do use landcover at sub-12-km resolution to calculate the 12-km predictor values for many of the spatial variables. For example, fractional forest coverage and forest biomass comes from our 1-km resolution forest-ecosystem model (DYNAFFOREST) and the fractional cover of other landcover types such as barren, water, and grass/shrub are calculated from 30-m maps. We have added a sentence in this vein to the end of the methods section 3.4 about landcover:

L343–345: "*Likewise, our use of sub-12-km landcover data to produce landcover predictors allows our modelling to include the effects of within-grid heterogeneity of fuel conditions, which is important given that most fires are smaller than 144 km².*"

In addition, we have added the following note to the paragraph about fire spread in the Discussion section:

Finally, the previous version of the Discussion did indicate how fire-spread is simulated within our forest-ecosystem model at the sub-12-km level as well as opportunities for more improvements of sub-grid processes. We have modified that text and it now reads:

L1009–1014: "*Another opportunity for improvement is to explicitly simulate fire spread. Currently, WULFFSS only estimates the final forest area burned by each simulated fire. When coupled within DYNAFFOREST, the ignition of a given simulated fire is assigned to a random 1-km forested-grid cell within the 12-km grid cell of WULFFSS and the fire spirals through adjoining or nearby forested areas until the pre-determined fire size is achieved or no nearby forested grid cells remain. Future improvements to WULFFSS should include estimating ignition location at sub-12-km resolution and modelling fire spread while maintaining computational efficiency.*"

2. The authors have considered a comprehensive list of potential predictors for constructing the Sp, Cp, and Tp components. However, the rationale or guiding principles for assigning specific variables to each of these three domains (Spatial, Annual-cycle, Temporal anomaly) is not sufficiently elaborated. A clearer explanation of the criteria used to categorize predictors into S, C, or T would significantly enhance the methodological transparency and reproducibility.
This comment is similar to the second comment from Reviewer #1 and we agree that the original submission fell short in describing the logic behind splitting the variables into the three components and we expanded the second paragraph of section 4 (Model description) to state:

L395–401: "*The S component is constructed first to capture the how variations in fire activity are driven by factors that are far more variable in space than in time, as these factors (e.g., forest biomass, lightning frequency, variables related to human population and fire suppression) are likely to modulate the sensitivity of fire activity to temporal variables. The C component is then constructed to account for variations in fire activity that are due to the mean annual climate cycle. Finally, the T component is constructed to account for effects of interannual climate variability, which are likely to be strongly modulated by the effects of the S and C variables already accounted for.*"

We also note that the original submission did describe the guiding principles in terms of how it was determined how it was determined which domain a given variable would be assigned to. In section 4, this is what was stated for each domain, now with slight revisions to clarify that the mean annual cycle (*C*) and temporal anomaly (*T*) components are composed of climate variables.

Spatial (*S*): L403–405: "*The S component represents drivers of forest-fire occurrence or size that are most variable in the spatial domain, such as topographic slope, fuel availability, human*

*population, mean annual lightning frequency, and long-term mean aridity, all of which may directly influence fire occurrence and also modulate the effects of C and T.*"

Mean annual cycle (*C*): L411–415: "*The C component represents climatological drivers of forest-fire occurrence or size that are most variable in the domain of the mean annual cycle, such as long-term means of each month's lightning frequency as well as variables that influence the seasonality of fuel moisture such as prec, solar, and VPD.*"

Temporal anomalies (*T*): L422–429: "*The T component represents climatological drivers of forest-fire occurrence or size that are most variable in the temporal domain of interannual and longer. ... Because T is meant to represent climate variability beyond the annual cycle, T variables are standardized so that for a given variable in a given grid cell, values have a mean of 0 and standard deviation of 1 for each of the 12 months during the calibration period.*"

3. The model commendably incorporates interaction terms between the S, C, and T composite predictors, but these are limited to pairwise (two-way) interactions. The potential three-way interaction (S×C×T) is not considered. Could the authors please justify this methodological choice?

This comment motivated us to produce an alternative version of the model that includes 3-way interactions between *S*, *C*, and *T* variables. Our choice to not do so originally was simply to limit the computational cost of parameterizing the fire-probability model as well as some worry that *SxCxT* interactions could produce occasional extreme simulation outcomes that are unrealistic. When we introduced the 3-way interactions in light of this reviewer comment, we found the performance of the new model was virtually identical to that of the old. Below in Fig. R2 we provide alternative versions of Figures 17 and 18, where we show how the new simulations that include 3-way interactions compare to observations in terms of annual and monthly fire frequencies and areas burned at the scales of the western US and regionally. Comparing these to Figures 17 and 18 in the resubmitted paper indicates that the alternative model with 3-way interactions performs virtually identically to, and not better than, the original model. Notably, we evaluated the behavior and performance of the alternative model with more depth than simply generating the alternative versions of Figs. 17 and 18 below. We found that allowing for 3-way interactions does generally lead to slightly more variables being included in the construction of the *S*, *C*, and *T* predictors, but these additional variables have minimal impact because the most impactful variables were consistently selected in the same order and with the same relationships to fire probability/size regardless of whether the 3-way interactions were included or excluded. Thus, we have decided to not include 3-way interactions in the final version of the model. We have added the following sentence to section 4.1 (Model framework):

L466–467: "*Notably, we considered including three-way interactions between the S, C, and T predictors in the P and A models but doing so did not improve model skill.*"

[Figure]

*Fig R2. Alternative versions of Fig. 17 (left) and Fig. 18. (right) where the probability and size models each include a 3-way interaction term between the spatial, seasonal, and temporal predictors. Comparison to Figs. 17 and 18 in the resubmitted paper indicates that model performance is minimally influenced, and not systematically positively, but inclusion of the additional interaction term.*

4. I recommend including a schematic diagram illustrating the general framework of WULFFSS. This figure should visually depict the relationships between the three core statistical models (P, N, A), their required input data streams (landcover, topography, climate, etc.), the key data processing steps (e.g., resampling), and the bidirectional coupling relationship between WULFFSS and the DYNAFFOREST model.

We agree that a diagram illustrating the general model framework would be helpful. We have added such a figure (now Figure 4 in the manuscript) and we provide it below as Figure R3.

[Figure]

*Figure R3: Flowchart outlining the general framework of the WULFFSS.*

Figure 3: Adding text labels (e.g., "Pacific Northwest" or "PNW") onto the map for the four quadrant regions would improve its immediacy and clarity.
We have added a legend to Figure 3 to clarify the names of the regions.

Figure 4: Providing quantitative goodness-of-fit metrics for the curve fits in each panel would be welcome. Furthermore, the relationship for some predictors (e.g., wetds_mean_1mo in panel b) appears weak or poorly captured by the fitted curve.
We now include the AICc and p-values associated with each curve fits shown in the figure showing P model predictors (now Fig. 5) as well as the figure showing A model predictors (now Fig. 8).

Line 37: "fire" to "fires"
We fixed this typo and thank the reviewer for pointing it out.